# Epistatic interactions between the high pathogenicity island and other iron uptake systems shape *Escherichia coli* extra-intestinal virulence

Guilhem Royer [1,2,3,4,5], Olivier Clermont[1], Julie Marin [1,6], Bénédicte Condamine [1], Sara Dion[1], François Blanquart[7], Marco Galardini [8,9] & Erick Denamur [1,10] ✉

The intrinsic virulence of extra-intestinal pathogenic *Escherichia coli* is associated with numerous chromosomal and/or plasmid-borne genes, encoding diverse functions such as adhesins, toxins, and iron capture systems. However, the respective contribution to virulence of those genes seems to depend on the genetic background and is poorly understood. Here, we analyze genomes of 232 strains of sequence type complex STc58 and show that virulence (quantified in a mouse model of sepsis) emerged in a sub-group of STc58 due to the presence of the siderophore-encoding high-pathogenicity island (HPI). When extending our genome-wide association study to 370 *Escherichia* strains, we show that full virulence is associated with the presence of the *aer* or *sit* operons, in addition to the HPI. The prevalence of these operons, their co-occurrence and their genomic location depend on strain phylogeny. Thus, selection of lineage-dependent specific associations of virulence-associated genes argues for strong epistatic interactions shaping the emergence of virulence in *E. coli*.

*Escherichia coli* extra-intestinal infections represent a considerable burden both in human and veterinary medicines[1,2]. *E. coli* is the first bacterial pathogen in humans responsible for deaths associated with antibiotic resistance[3]. The population structure of *E. coli* is globally clonal[4] with the delineation of several phylogenetic groups and numerous sequence types[5]. The virulence of the strains is mainly due to the presence of virulence associated genes (VAGs) present on the chromosome and/or plasmids and coding for adhesins, protectins, iron capture systems, invasins and toxins. As *E. coli* acts as an opportunistic extra-intestinal pathogen[6], the severity of the disease is mainly due to host characteristics as the age, the presence of comorbidities and to the portal of entry for bloodstream infections[7,8]. In this context, intrinsic extra-intestinal virulence of the strains is often assessed in animal models. The chicken systemic

[1]Université Paris Cité, IAME, INSERM, Paris, France. [2]Département de Prévention, Diagnostic et Traitement des Infections, Hôpital Henri Mondor, Créteil, France. [3]LABGeM, Génomique Métabolique, Genoscope, Institut François Jacob, CEA, CNRS, Université Paris-Saclay, Evry, France. [4]EERA Unit "Ecology and Evolution of Antibiotics Resistance," Institut Pasteur-Assistance Publique/Hôpitaux de Paris-Université Paris-Saclay, Paris, France. [5]UMR CNRS, 3525 Paris, France. [6]Université Sorbonne Paris Nord, IAME, INSERM, Bobigny, France. [7]Center for Interdisciplinary Research in Biology, CNRS, Collège de France, PSL Research University, Paris, France. [8]Institute for Molecular Bacteriology, TWINCORE Centre for Experimental and Clinical Infection Research, a joint venture between the Hannover Medical School (MHH) and the Helmholtz Centre for Infection Research (HZI), Hannover, Germany. [9]Cluster of Excellence RESIST (EXC 2155), Hannover Medical School (MHH), Hannover, Germany. [10]AP-HP, Hôpital Bichat, Laboratoire de Génétique Moléculaire, Paris, France. ✉e-mail: erick.denamur@inserm.fr

infection model via air sacs[9] and the mouse sepsis assay[10,11] are robust, reproducible and widely used models.

Using the mouse model, it has been shown that the effects of these VAGs on virulence are cumulative[12] and dependent on the genetic background of the strain[13,14], an argument for epistatic interactions between loci across the genome. Extra-intestinal pathogenic *E. coli* (ExPEC) with multiple VAGs belong mainly to phylogroups B2 and D and are highly virulent in the mouse model[10]. Recently, using this assay coupled to a genome wide association study (GWAS) in a collection of 370 strains representative of the genetic diversity of the *Escherichia* genus, the iron capture systems were shown to have a major role in virulence with the high-pathogenicity island (HPI) having the greatest association, followed by the aerobactin (*aer*) (*iuc/iut* genes) and *sit* operons[15]. Interestingly, the aerobactin siderophore system as well as the Sit iron transport system can be encoded by Col-like plasmid genes and/or chromosomal genes[16] whereas the HPI is a chromosomal genomic island which encodes for a siderophore (yersiniabactin) mediated iron-uptake system[17]. Salmochelin, another siderophore found in ExPEC, is encoded by the *iro* gene cluster located either on Col-like plasmids and/or on chromosomal islands[18].

Although most of the extra-intestinal pathogenic clones belong to the phylogroups B2, D and to a lesser extend C and F[6], a lineage responsible for human extra-intestinal infections and antibiotic resistance belonging to the phylogroup B1 has recently emerged, namely the ST58[19] according to the Warwick University nomenclature, belonging to the clonal complex (CC)87 according to the Institut Pasteur nomenclature[20]. Using phylogenomic comparative analysis, a role of ColV plasmid, together with the HPI, has been pointed in the evolution of the pathogenicity of this clone[19].

Thus, a clear picture of the respective role of the VAGs in the emergence of virulence, according to their genomic location and to the phylogenetic background of the strains, is lacking. In the present work, we deciphered the role of the ColV plasmid and HPI in the CC87 virulence using GWAS based on the mouse assay phenotype. We then extended the study of the role of the various iron capture systems to the species as a whole. Our goal was to understand how functional redundancy, epistatic interaction and genome location are acting at the species level.

## Results

### The CC87 is composed of five subgroups, one of them exhibiting numerous VAGs and antibiotic resistance genes

We first reconstructed the phylogenetic history of both ST58 and its sister group ST155 which are parts of the CC87 by analyzing the core SNPs of 232 strains, including 26 ST58 strains representative of the diversity previously described by Reid et al.[19] (Fig. 1, Supplementary Data 1). These strains were recovered from humans ($n = 125$), animals ($n = 87$) as well as the environment ($n = 20$) and were diverse in terms of geographical origin (Europe $n = 91$, America $n = 59$, Australia $n = 50$, Africa $n = 30$, and Asia $n = 2$).

By rooting the tree on the B1 phylogroup ST1128 IAI1 strain, and based on both Warwick University (WU) and Institut Pasteur (IP) multi-locus sequence typing (MLST) schemes, five main subgroups were distinguished (Fig. 1). The most basal one is the ST$^{WU}$58-ST$^{IP}$186 (ST58/186) subgroup ($n = 7$), followed by the ST58$^{WU}$-ST$^{IP}$87-A (ST58/87A) ($n = 12$), ST$^{WU}$155-ST$^{IP}$21 (ST155/21) ($n = 93$) and ST$^{WU}$58- ST$^{IP}$87-B (ST58/87B) ($n = 57$) subgroups, and the more recently emerged ST$^{WU}$58-ST$^{IP}$24 (ST58/24) (n = 63) subgroup. The small discrepancies between Reid et al. clustering and our subgroup definition are probably due to differences in (i) the clustering approaches (fastbaps model-based clustering vs combination of phylogeny and multi-locus sequence typing), (ii) the alignment used to compute the phylogenetic tree (core gene alignment without filtering for recombination vs recombination-free core genome alignment) (Fig. S1) and (iii) the strain sampling (ST58 *sensu stricto* versus CC87). Importantly, the ST58/24 encompasses the BAP2 cluster described in[19].

We then screened our collection for VAGs and antimicrobial resistance genes (ARGs). As reported previously[19], the ST58/24 subgroup exhibited significantly more VAGs classified in toxin, protectin and iron acquisition system categories (Fig. 2a, Supplementary Data 2) and was predicted to be more antibiotic (ampicillin and trimethoprim) resistant (Fig. S2, Supplementary Data 3) than the other subgroups. Moreover, almost all strains in this subgroup carry the HPI and the ColV plasmids. Of note, whereas this subgroup represented 27% of our collection, it encompassed two-third of the strains isolated from human extra-intestinal infections (Fig. S3), an over-representation also observed by Reid et al.[19].

Altogether, these first analyses confirmed that virulence and resistance genes within the *E. coli* B1 phylogroup ST58/ST155 clonal complex (CC87) are mainly present in the ST58/24 lineage.

### Mouse sepsis assay coupled to GWAS identifies the HPI as the major driver of virulence within the CC87

In a second step, we tested a set of 70 strains (30% of our data set) distributed in the five main subgroups in the mouse model of sepsis (Fig. 1). Twenty-four strains have been tested previously[15] whereas the 46 remaining were tested in the present work (Supplementary Data 1).

Mouse survival curves showed that the strains exhibited different levels of virulence according to their subgroup (Fig. 2b). The ST58/24 subgroup killed more than all CC87 subgroups except the ST58/87A, in accordance with the VAG content. The ST58/186 and ST155/21 killed less than the other CC87 subgroups, ST58/186 even behaving as the K-12 avirulent control strain. Interestingly, the virulence observed for the B1 phylogroup strains is significantly lower than the control B2 phylogroup CFT073 strain, a potential effect of the role of the genetic background of the strains on the expression of virulence[13].

To identify the genetic determinants responsible for the mouse virulence phenotype, we performed a GWAS with Pyseer[21] using as input the unitigs calculated from the genome assemblies or the gene presence/absence and the number of mice killed by each strain. As shown on the volcano plot (Fig. 2c), 417 and 16 unitigs were positively (in red) and negatively (in blue) significantly associated with the phenotype, respectively. All these unitigs except one correspond to the HPI insertion region (Fig. 2d). The only non-HPI-related unitig was associated with a gene encoding a phage minor tail protein. Moreover, gene presence/absence also points to the HPI with 12 genes over the 13 significantly associated with virulence related to this virulence determinant (Fig. 2d). In contrast, no unitig or gene related to the ColV plasmids were significantly associated with virulence (Fig. 2c).

### At the species level, both *aer* and *sit* operons, but not the plasmid by itself, are involved in the mouse sepsis phenotype

We then wanted to evaluate the role of the VAGs used to infer the presence of ColV plasmids in extra-intestinal virulence, but this time at the species level. For this purpose, we used the data of a GWAS analysis performed on 370 strains of the genus *Escherichia* that have been tested in the mouse sepsis assay, including 24 strains of the CC87 (see above). In this work, the HPI was the strongest genetic element associated with virulence in mice, followed by the aerobactin (*iucABCD/iutA*) and *sit* (*sitABCD*) operons[15].

We determined the genomic location (i.e. chromosomal or plasmidic) of the VAGs with PlaScope[22]. From this analysis, we were able to show that some genes were strictly plasmidic (*cvaC*, *cvi*, *etsABC*, *hlyF*, *ompT*) whereas others could be located on both elements. The VAGs *iucABCD* and *sitABCD* were predominantly chromosomal in 53.96–54.35% and 72.73–73.27% of the cases, respectively. The distribution of VAGs also differed according to the phylogeny (Fig. 3). We found most of the VAGs in phylogroups B1, B2, C, D, F and G at a variable level. They were predicted to be almost exclusively located on

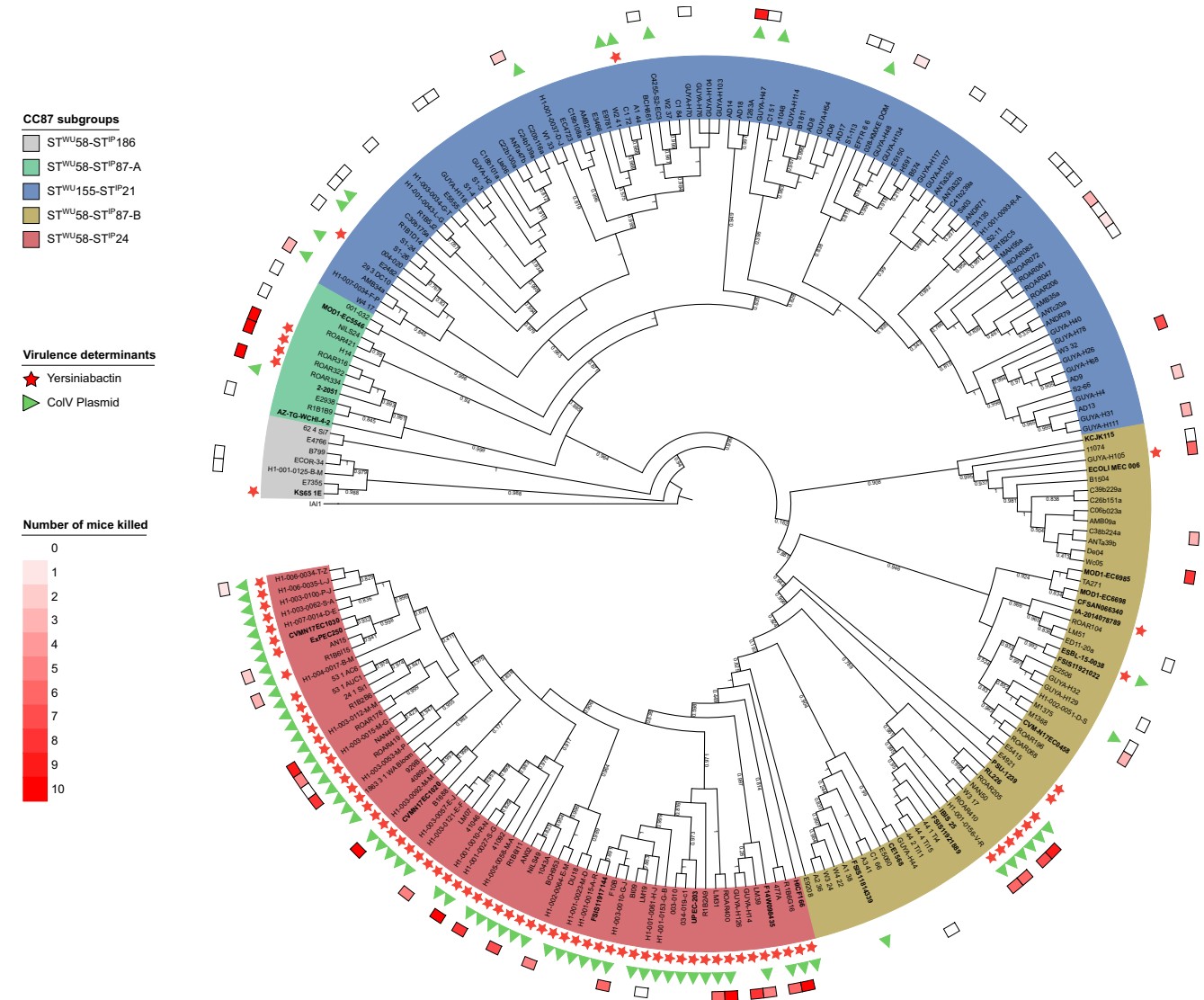

**Fig. 1 | Maximum likelihood core genome phylogenetic tree of the 232 B1 phylogroup CC87 (Institut Pasteur scheme numbering) strains.** The five CC87 subgroups (ST$^{WU}$58-ST$^{IP}$186, ST$^{WU}$58-ST$^{IP}$87-A, ST$^{WU}$155-ST$^{IP}$21, ST$^{WU}$58-ST$^{IP}$87-B, ST$^{WU}$58-ST$^{IP}$24) based on the Warwick University and Institut Pasteur MLST schemes are highlighted in color. The presence of the high pathogenicity island (HPI) is highlighted by red stars and ColV plasmid (defined as proposed by Reid et al.[19]) by green triangles. In the outermost circle, colored squares represent the number of mice killed over ten in the mouse model of sepsis[11]. The 26 ST58 genomes obtained from the study by Reid et al. are in bold. The tree is rooted on IAI1 (non-CC87, phylogroup B1). For the sake of readability, branch lengths are ignored and local support values higher than 0.7 are shown.

plasmids in phylogroups B1, C and G, with a prevalence over 40% in the latter two phylogroups. At the opposite, chromosomal location was mainly observed in the archetypal ExPEC phylogroups B2, D and F. We ran the same analysis on the 232 genomes of the CC87 and confirmed the highest prevalence (70%) of VAGs in the ST58/24 subgroup and the almost exclusive plasmidic location of the VAGs in this population (Fig. S4).

Then, using GWAS, we looked at associations of unitigs belonging to each of these VAGs with virulence in mice. Interestingly, we found significant associations mainly for the aerobactin and *sit* operons (Fig. 3). The sequences of VAGs do not significantly differ depending on their predicted location (Fig. S5) and when analyzing the association results according to the predicted location of each VAG in each strain, we did not observe appreciable differences according to genomic location (Fig. S6). When focusing on genes that were exclusively located on a plasmid, such as *hlyF* and *ompT*, we found that all unitigs mapped to them were far below the significant threshold (Fig. S6). If we consider the presence of the ColV plasmid as a covariate

in the GWAS analysis, the unitigs associated with *iucABCD/iutA* and *sitABCD* remain significant, arguing for a role beyond their simple location on the plasmid (Fig. S7).

Overall, these data indicate that within the *Escherichia* genus, the sole presence of the ColV plasmid does not explain virulence in our mouse sepsis model, even though aerobactin and *sit* operons can be found as plasmid borne.

### The HPI needs other VAGs to express full virulence

Next, we assessed the respective role of the iron acquisition-related operons *aer*, *sit* and *iro* as compared to the higher predictor of mouse death, the HPI, in the same 370 strains' dataset. We found high odds ratios for the status "killer" (i.e. at least 9/10 mice killed per strain)[11] as a function of the presence of the HPI in strains carrying different VAG combinations. Indeed, we found a significant association of the HPI with the status "killer" for all combinations of accompanying VAGs, except strains with *iroN* alone, *iucA* alone and the combination *iucA/iroN* (Fig. 4). However, the HPI was associated with death of at

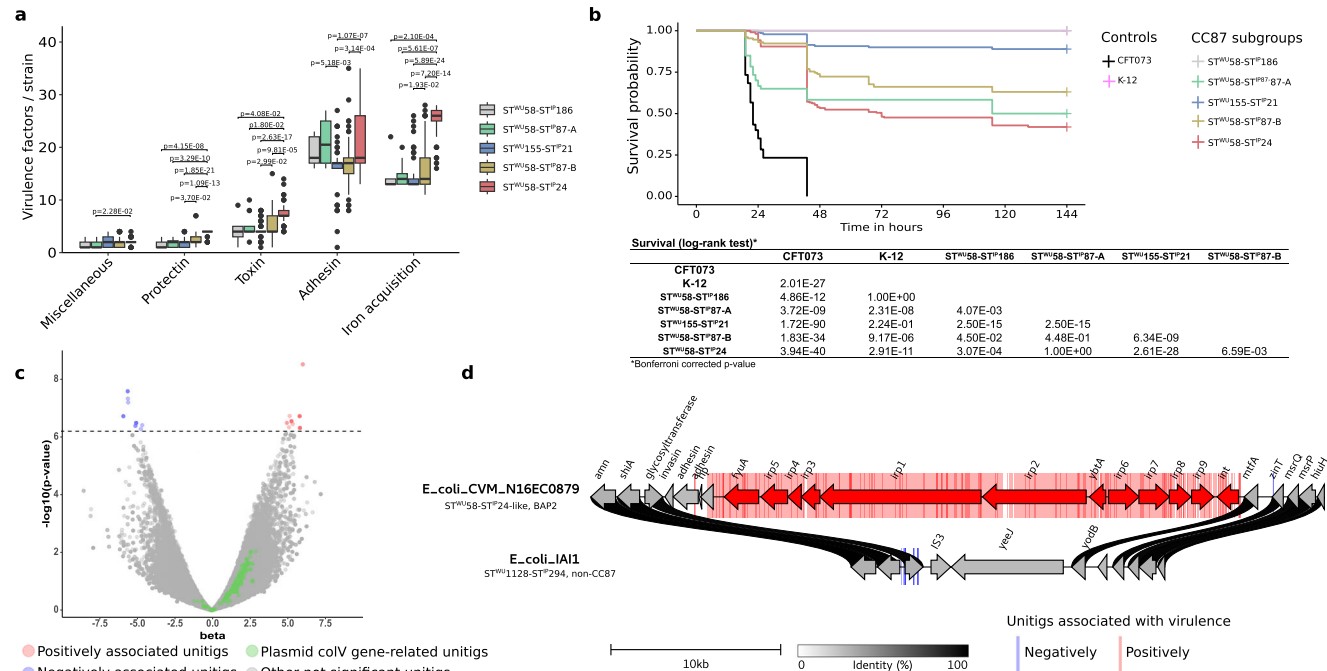

**Fig. 2 | From genotypic and phenotypic characterization of CC87 extra-intestinal virulence to identification of genetic determinants by GWAS in the 232 B1 phylogroup CC87 strains. a** Number of virulence-associated genes per strain among the six main functional classes of virulence according to CC87 subgroups (*n* = 232 biologically independent CC87 genomes) (Kruskal-Wallis test with Bonferroni adjusted *p*-values). Only significant *p*-values are shown. The upper and lower limits of box-plots represent 75th and 25th quartile, the centre line represents the median and the whiskers extend to 1.5 × IQR. Dots represent values outside these ranges. **b** Kaplan-Meier survival curves of the mouse sepsis assay. Note that the pink (K-12) and gray (ST^WU58-ST^IP186) curves overlap. The table below the curves shows the log rank results to compare survival according to CC87 subgroups. **c** Results of the unitigs association with virulence (likelihood ratio test). The *p*-value of the association is shown on the *y*-axis, the effect size (beta) on

the *x*-axis and the significance level with a dotted line (Bonferroni multiple-testing corrected *p*-value). The unitigs positively and negatively associated with the phenotype are highlighted in red and blue, respectively. The unitigs found in genes belonging to the ColV plasmid (as described by Reid et al.[19]) are highlighted in green. Other non-significant unitigs are in gray. **d** Physical map of the genome region where significant associations with the virulence in mice were observed. Unitigs positively or negatively associated are represented in the background of the map by red and blue bars, respectively. Genes positively associated with virulence are represented by red arrows and include the whole HPI. The fully sequenced and circularized genomes of *E. coli* CVM_N16EC0879 (ST^WU58-ST^IP24-like, BAP2 in the study by Reid et al.[19]) and IAI1 (ST^WU1128-ST^IP294, non-CC87) were used as reference. The links between the maps are colored according to amino acid identity.

least 5/10 mice per strain in only 50% of the cases when it was alone without *iroN*, *iucA* and *sitA*. The strains not killing or killing only one mouse despite the presence of the HPI belonged to phylogroups A (*n* = 5/9), B2 (*n* = 2/9), C (*n* = 2/9) (Fig. S8).

Overall, these data indicate that, in most cases, the presence of the HPI is necessary but not always sufficient for a strain to be virulent, with other iron uptake systems required to express full virulence. Conversely, these accessory systems alone are not sufficient to confer full virulence.

## The HPI co-occurs with *aer, sit* and *iro* operons in *E. coli* genomes at different frequencies depending on the phylogeny of the strain

To get a broader view of VAG co-occurrences and their genomic location, we used the 2302 high-quality circularized *E. coli* genomes available in RefSeq (https://www.ncbi.nlm.nih.gov/refseq/). Using this dataset prevented us from the potential bias resulting from the Illumina short-read sequencing when we analysed the genomic location of VAGs among the 370 *Escherichia* strains. Nonetheless, we confirmed that PlaScope enables a classification of chromosomal and plasmidic contigs with high performances (Fig. S9).

First, we focused on the five main ST responsible for bacteremia in France[23] [ST131 (*n* = 123), ST73 (*n* = 41), ST69 (*n* = 33), ST95 (*n* = 46), ST10 (*n* = 48)] as well as the STc58 (*n* = 37) to quantify co-occurrence frequencies and prevalence of VAG pairs. In all these ST/STc, with the exception of ST73 in which they are not found, the plasmidic genes co-occurred at a similar prevalence (from a few percent in ST10 and ST131

to 75% in ST95). We measured their co-occurrence by computing odds ratios based on a 2 × 2 contingency table from the presence-absence of each pair of genes to be compared. We observed very high odds ratios (Supplementary Data 4), which supports their frequent co-location (Fig. 5). More precisely, we observed distinct patterns in different STs. In ST131, the overall pattern is dominated by a very high prevalence and frequent co-occurrence of chromosomal VAGs, with a notable absence of *iroN* and the presence of *fyuA* in all genomes. Furthermore, the plasmidic VAGs are almost all negatively associated with the chromosomal *sitA* and *iucA*. The chromosomal dominant pattern is exacerbated in the ST73 with an almost complete fixation of VAGs on the chromosome. In ST69, odds ratios significantly greater than one (at the 0.05 level) are only found between VAGs sharing the same location. In this ST, we also noticed the absence of chromosomal *iroN*. In ST95, the pattern was somewhat different, with significant positive association between plasmidic VAGs and negative association between plasmidic VAGs and chromosomal *iroN*. These data are in agreement with the frequent presence of the ColV plasmids in the ST95[24]. Of note, *fyuA* was found in all genomes and chromosomal *iucA* was absent in this ST. At the opposite and consistent with its commensal behaviour[25,26], the ST10 carries few VAGs, which are usually clustered on plasmids or on the chromosome. Finally, for the STc58, in accordance with our previous observation and data from Reid et al.[19], *iroN*, *iucA* and *sitA* co-occurred mainly and frequently on plasmids (odds ratios range from 25.28 to infinite, *i.e.* perfect association) and are also in positive association with the HPI (odds ratios range from 15.74 to infinite).

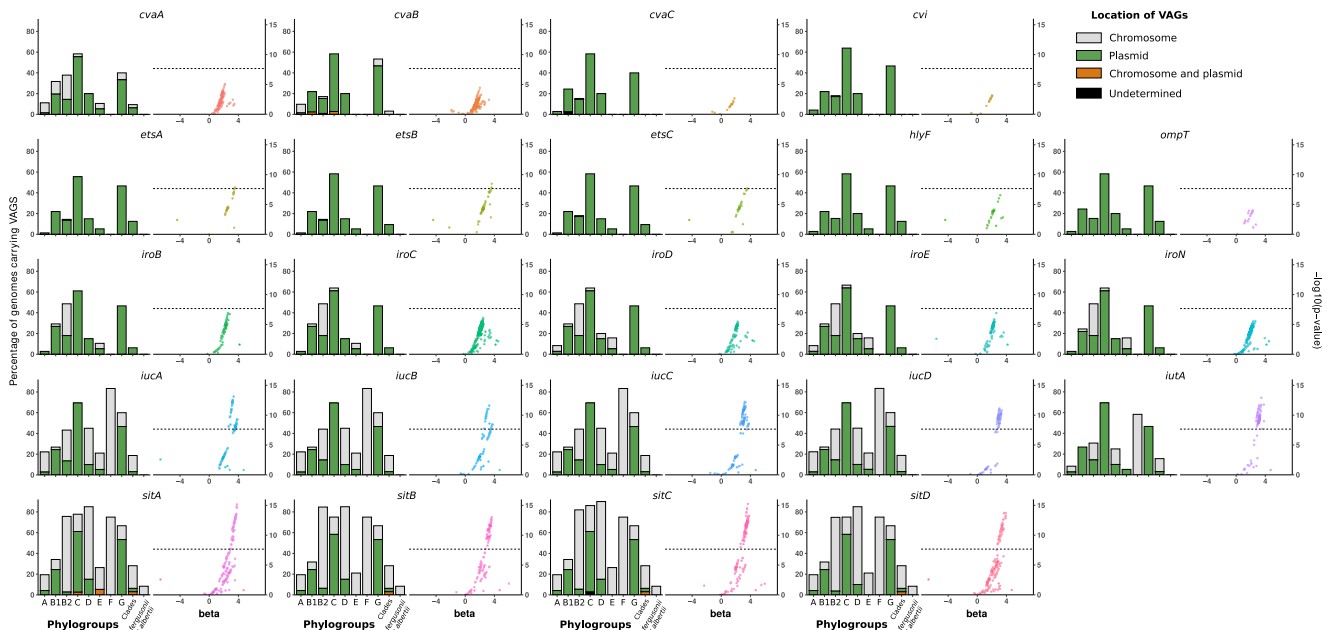

**Fig. 3 | Prevalence and location of VAGs and association between virulence and unitigs within these VAGs among 370 genomes of the *Escherichia* genus.** For each VAG two plots are represented. The bar plot represents the prevalence according to the phylogroup/genus and the predicted location. Plasmidic location is highlighted in green, chromosomal location in gray and cases with both chromosomal and plasmidic location in orange. Location of VAGs could not be determined in two genomes (*cvaC* in phylogroup B1 and *sitC* in phylogroup C) and is shown in black. The dataset is composed of genomes of strains from phylogroup A ($n = 72$), B1 ($n = 41$), B2 ($n = 111$), C ($n = 36$), D ($n = 20$), E ($n = 19$), F ($n = 12$), G ($n = 15$), clades ($n = 32$) and *E. fergusonii* and *E. albertii* ($n = 12$). The scatter plot represents the association between unitigs within the VAGs and the virulence in mice (likelihood ratio test). The *p*-value is shown on the *y*-axis, the effect size (beta) on the *x*-axis and the significance level of the GWAS analysis with a dotted line (Bonferroni multiple-testing corrected *p*-value).

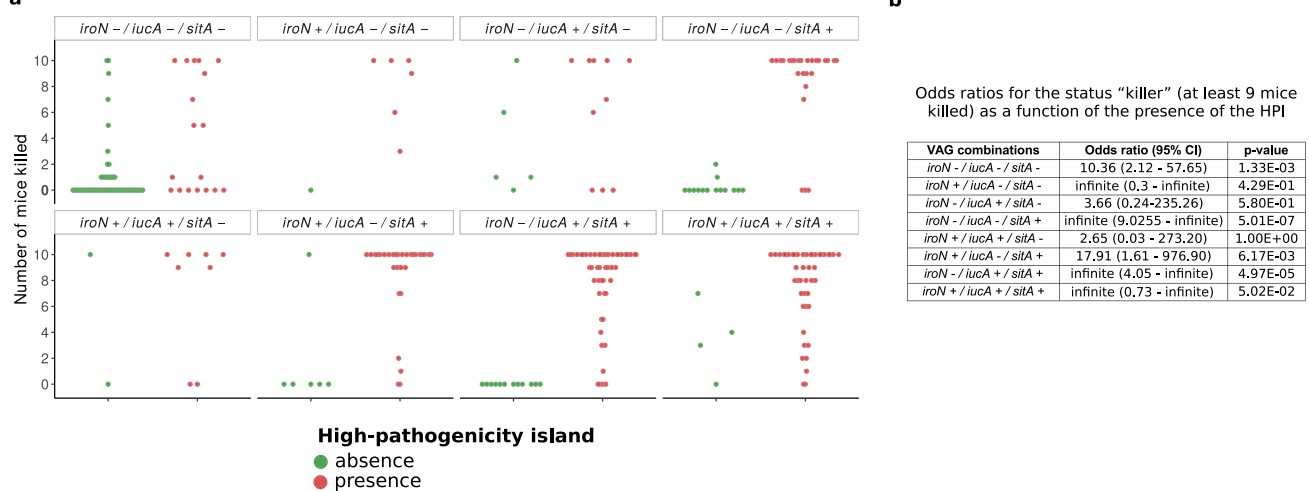

Odds ratios for the status "killer" (at least 9 mice killed) as a function of the presence of the HPI

| VAG combinations | Odds ratio (95% CI) | p-value |
|---|---|---|
| *iroN* - / *iucA* - / *sitA* - | 10.36 (2.12 - 57.65) | 1.33E-03 |
| *iroN* + / *iucA* - / *sitA* - | infinite (0.3 - infinite) | 4.29E-01 |
| *iroN* - / *iucA* + / *sitA* - | 3.66 (0.24-235.26) | 5.80E-01 |
| *iroN* - / *iucA* - / *sitA* + | infinite (9.0255 - infinite) | 5.01E-07 |
| *iroN* + / *iucA* + / *sitA* - | 2.65 (0.03 - 273.20) | 1.00E+00 |
| *iroN* + / *iucA* - / *sitA* + | 17.91 (1.61 - 976.90) | 6.17E-03 |
| *iroN* - / *iucA* + / *sitA* + | infinite (4.05 - infinite) | 4.97E-05 |
| *iroN* + / *iucA* + / *sitA* + | infinite (0.73 - infinite) | 5.02E-02 |

**Fig. 4 | Virulence according to VAG combinations and HPI. a** Number of mice killed over ten according to the combination of iron acquisition-related VAGs *iroN*, *iucA*, *sitA*. These individual genes are representative of their gene cluster. In each facet, each point represents the number of mice killed by a given strain according to the VAG or combination of VAGs it carries. Strains with or without the HPI are represented by red and green points, respectively. **b** Odds ratios (95% confidence interval) and *p*-value for the status "killer" (i.e. at least 9/10 mice killed)[11] (Two-sided Fisher exact test) as a function of the presence of the HPI in strains carrying different VAG combinations. A related figure detailing the phylogroup of each strain is available in Fig. S8.

Then, looking at the broader scale of phylogroups, the co-occurrences are also phylogroup-specific, with mainly three types of patterns (Fig. S10). In phylogroups A, B1 and E, the prevalence of VAGs are overall low. Where significant positive associations are identified, they involve either only plasmidic VAGs, only chromosomal VAGs, or *fyuA* with chromosomal or plasmidic VAGs. In phylogroups C, F and G, the plasmidic VAGs are more prevalent and all in significant positive associations (odds ratios range from 10.84 to infinite). Other significant associations were found for plasmidic *iucA* and *sitA*, positively associated with *fyuA* in phylogroups C and G and negatively with chromosomal *iucA* and *sitA* in phylogroup F. Some other chromosomal VAGs are found significantly associated in phylogroups C (*iroN* with *sitA* and *fyuA*) and F (*sitA* with *iucA*). Finally, among phylogroups B2 and D, chromosomal VAGs are highly prevalent and significantly and positively associated with each other (except *iroN* which is absent from phylogroup D). Again, the plasmidic VAGs are also associated with each other, but in phylogroup B2 they are negatively associated with chromosomal *iroN* and *iucA*.

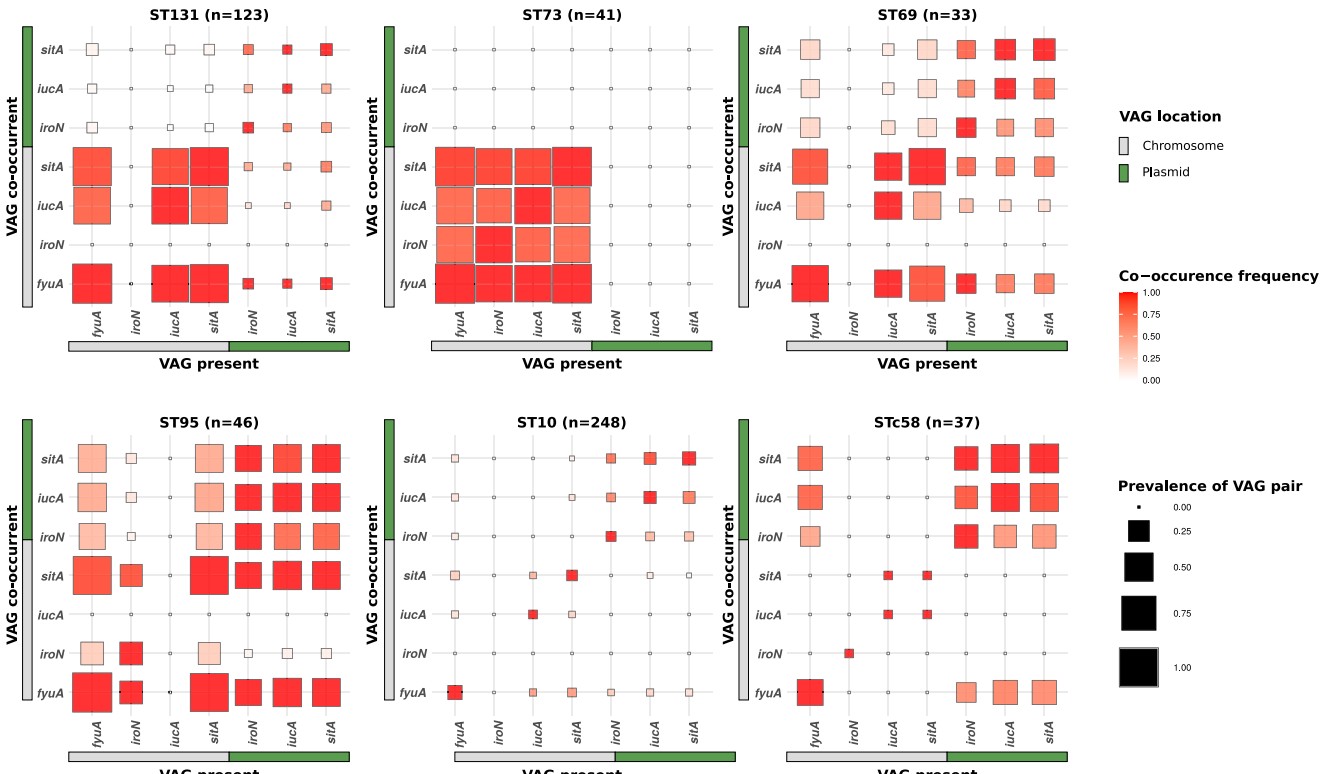

**Fig. 5 | Co-occurrence frequency and prevalence of iron acquisition-related VAG pairs among 528 fully circularized *E. coli* genomes belonging to ST131, 73, 69, 95, 10 and STc58.** For each given VAG on the *x*-axis, the frequency of co-occurrence with the VAGs on the *y*-axis is highlighted by a colour gradient. The size of each square is proportional to the prevalence of the VAG pair in the given ST/STc.

VAGs are separated according to their location on the chromosome in gray or on the plasmid in green. The co-occurrence frequencies of the VAGs and the prevalence at the phylogroup scale are available in Fig. S10. Odds ratios to test for associations between chromosomal and/or plasmidic VAGs in a given ST/STc are available in Supplementary Data 4.

Overall, the co-occurrences of VAGs show patterns that are specific to ST/STc and/or phylogroups. Frequent co-occurrences and high prevalences of chromosomal VAGs are mainly observed in typical ExPEC like ST131, 73, 95, 69 or more broadly in phylogroups B2 and D. Of note, *iro* or *iuc* may be absent from the chromosome depending on the clone. Interestingly, in non-typical ExPEC (STc58, A, B1, C, E, G) plasmidic *iro*, *aer* and/or *sit* operons are positively associated with *fyuA*.

**The VAG co-occurrences result from selection**

To determine whether these co-occurrences were the result of "chance and timing" with a single gain followed by phylogenetic inertia or selection with multiple independent gains through the phylogenetic history indicating evolutionary convergence, we searched for the sites of chromosomal insertions of the genes and for incongruences between gene and strain phylogenies.

Whereas the HPI was mostly inserted at a single site but spread by homologous recombination within the *E. coli* species[17], *iro*, *sit* and *iuc* genes were inserted in multiple chromosomal sites (6, 14 and 10, respectively) (Supplementary Data 5, 6, 7). These sites were mainly ST specific among a given phylogroup. As an example, in strains from ST127 (B2 phylogroup) *iro* gene cluster is inserted into the tRNA-SerW whereas in other B2 strains the insertion site is almost exclusively in the tRNA-SerX (Supplementary Data 5). However, STs belonging to different phylogroups sometimes shared the same site. Moreover, chromosomal co-location of VAGs on the same region of plasticity[27] was very rarely observed (only 13 cases of *iucA* and *sitA* co-location).

We compared the phylogenetic histories of *aer*, *sit* and *iro* based on nucleic sequence alignments of operons for each location (i.e. chromosome or plasmid) to the strain phylogeny to detect multiple horizontal gene transfer events (Fig. S11). When comparing the

phylogeny at the phylogroup level, a very high level of incongruence was noted for the plasmidic genes (Fig. S11b, d, f) and to a lesser extent for the chromosomal genes (Fig. S11a, c, e). We then compared patristic distances between all possible pairs of genomes according to their location and ST and/or phylogroup (Fig. 6). The patristic distances between plasmid operons were in all cases very small, reflecting their recent evolutionary origin compared to the species (Fig. 6b, d, f). The high level of incongruence and the small diversity together suggest high mobility of the plasmids. Chromosomal sequences exhibit on average larger patristic distances in different ST and/or phylogroup than within ST. However, some operon sequences from strains of different STs or phylogroups show patristic distances as small as between strains of the same ST (Fig. 6a, c, e): this suggests some degree of recent mobility of the chromosomal sequences. By running a phylogenetic analysis combining both plasmid and chromosomal operon sequences (Figs. S12, S13 and S14), we confirmed that sequences clustered according to their location. But in a few cases, we were able to identify chromosomal sequences nested within clades with a majority of plasmid sequences, and vice versa, for *iro*, *aer* and *sit*. This is in line with the identification of a chromosomal *iro* gene cluster on a transmissible plasmid in a clinical isolate[28].

Lastly, at the species level, we found significant associations between the presence of the HPI and *aer*, *sit* and *iro* genes whatever their genome location, using Pagel's model for correlated binary-trait evolution on a phylogenetic tree[29] (Supplementary Data 8). This represents an argument for traits evolving in response to selection[30].

In sum, we found recent multiple gains of VAGs carried by plasmids and, to a lesser extent, at chromosomal sites within the species phylogenetic history, a hallmark of selection. In some cases, we were even able to identify probable exchanges between these different genetic supports.

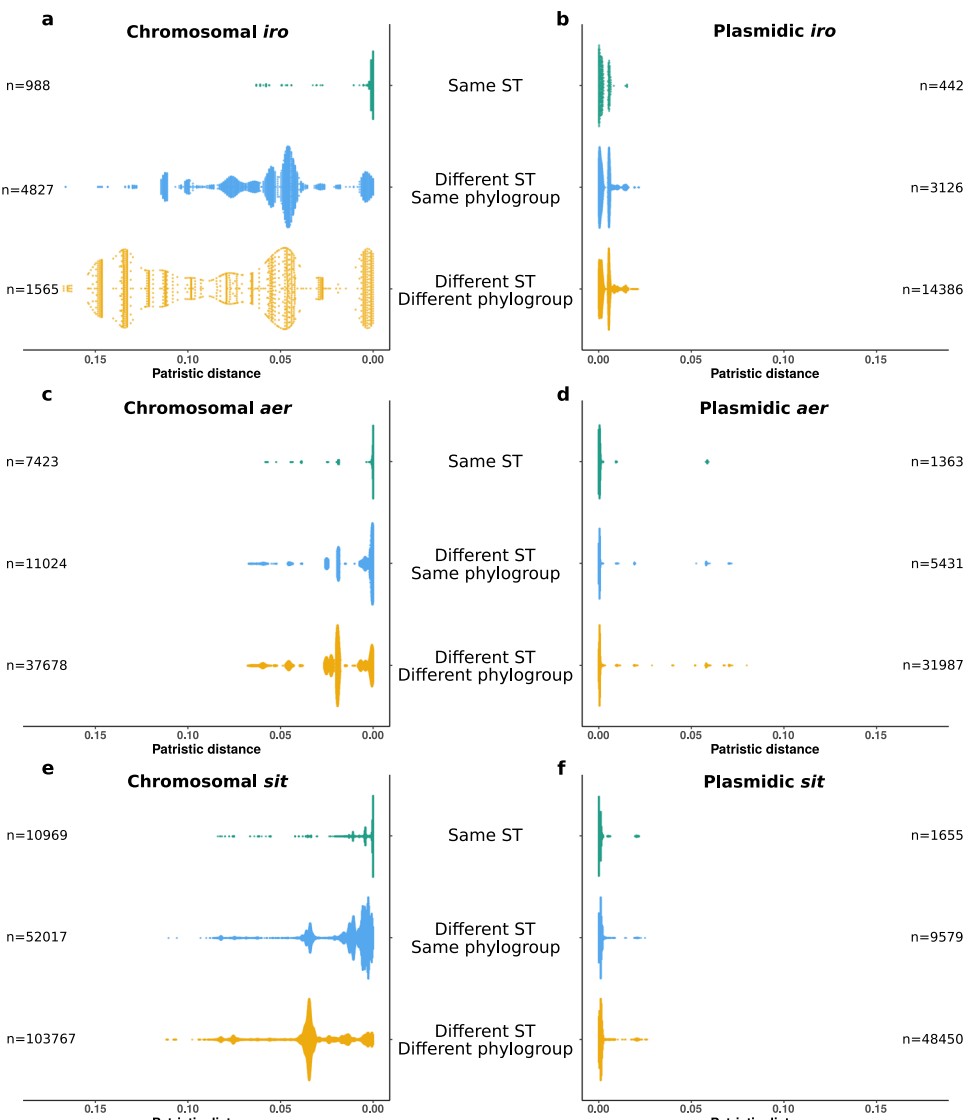

**Fig. 6 | Distributions of patristic distances between *iro*, *aer* and *sit* operons according to their location and to the sequence types and phylogroups of the strains.** The scatter plots represent the patristic distances between pairs of (**a**) chromosomal *iro* operons, (**b**) plasmidic *iro* operons, (**c**) chromosomal *aer* operons, (**d**) plasmidic *aer* operons, (**e**) chromosomal *sit* operons and (**f**) plasmidic *sit* operons. The number of pairs of genomes involved in each category is specified opposite each scatter plot. For the sake of readability, the highly divergent sequences of chromosomal *aer* from genomes GCF_010725305.1 and GCF_002946715.1 and chromosomal *sit* from genome GCF_024225755.1 were not included.

## HPI gene inactivation is associated with the presence of chromosomal *iro* and *sit* genes

It has been reported in some archetypal ExPEC strains as CFT073 that the absence of detectable yersiniabactin is due to numerous in-frame stop codons in the *irp1* and *irp2* genes[31]. Furthermore, within-host evolution of bacterial pathogens selects for loss of function of genes related to uptake or synthesis of siderophores[32] and social interactions of bacteria in community lead to the emergence of cheaters inactivating diffusible siderophore synthesis genes while benefiting from other cells' siderophore thanks to the presence of the receptor[33]. In this context, we wanted to know if some of the patterns observed above could be explained by such gene inactivation.

All inactivating mutations (nonsense, small and large deletion and insertion) were identified within the coding regions of the genes of the four iron capture systems in the 2302 RefSeq genomes (Supplementary Data 9). As a control, we used the MLST genes known to be under purifying selection except *uidA* which is known to be inactivated in numerous *E. coli*, especially the O157:H7 enterohemorrhagic

strains[34,35]. We computed the inactivation rate for each gene as the number of strains with at least one inactivation event over the number of strains carrying the gene, normalised by gene length (Fig. 7 and Supplementary Data 9). As expected, we noted a high rate for *uidA* and removed it for further analyses. By fitting a linear model to explain differences in rates, we detected a difference in inactivation rates between gene categories (*p*-value = 1.35e-04). MLST genes had the lower rate whereas *iro* operon genes had the higher rate, HPI, *aer* and *sit* operon genes being intermediate (Supplementary Data 9). Among iron capture systems, we did not find any association between the corrected inactivation rate and the gene function (*p*-value = 2.85e-01). Finally, when looking at individual genes, we found a high rate of inactivation for two biosynthesis genes, *irp2* and *iucB* (CI without overlap).

We then searched for associations between inactivation of genes in one system and presence of another system. We found significant associations only between inactivation of the HPI and the presence of chromosomal *iro* and/or *sit* genes, after correcting for phylogenetic structure (Pagel's model) (Supplementary Data 10).

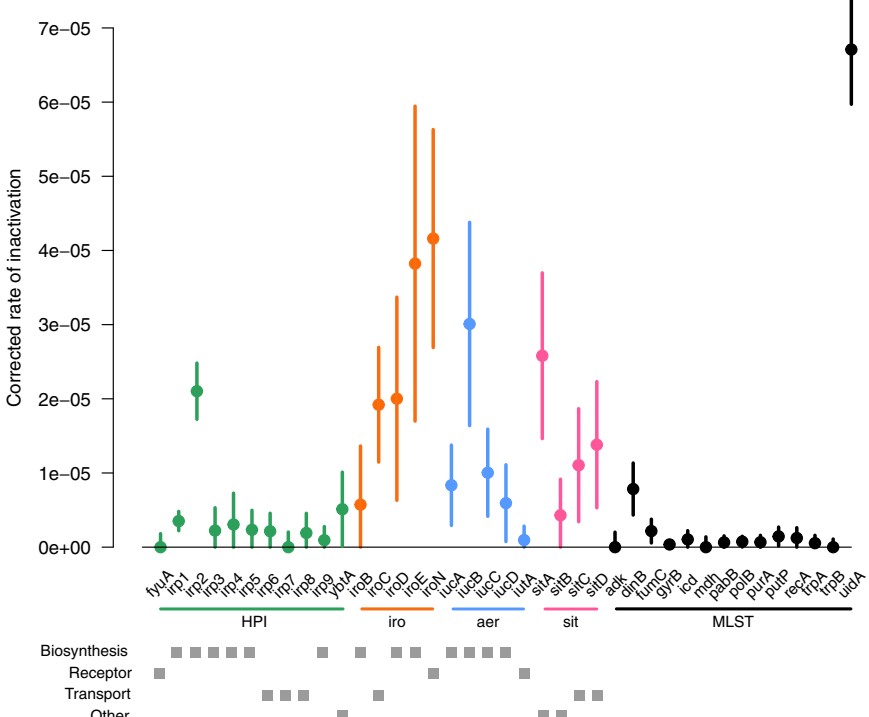

**Fig. 7 | Corrected inactivation rate of HPI (green), *iro* (orange), *aer* (blue) and *sit* (pink) operon genes and of MLST control genes (black) (*n* = 2302 biologically independent *E. coli* genome).** For each gene we show the corrected inactivation rate (circles at the centre of error bars) and the 95% binomial confidence interval (error bars) computed by approximating the distribution of error with a normal distribution, and using the rule of three for rates equal to 0. The inactivation rate is the number of strains with at least one inactivation over the number of strains carrying the gene and gene length. The gray squares indicate the gene function category (biosynthesis, receptor, transport and other). Note the high inactivation of the *uidA* gene previously reported[34,35].

Thus, iron capture system genes are inactivated at a higher pace than housekeeping genes and only the inactivation of HPI genes is associated with the presence of other systems encoded by the chromosome.

## Discussion

Classically, the confirmation of the role of a VAG in virulence is obtained by inactivating the gene and by showing that the KO mutant is no longer virulent or is outcompeted by the wild type strain in co-infection assays. Ideally, the complementation of the mutant should restore the wild-type phenotype. Such an approach has proven to be very powerful but it is particularly time consuming and usually only allows for the testing of single or double mutants[36–40]. Alternatively, deletion of pathogenicity islands (PAIs) allows to test simultaneously blocks of hundreds of genes[41,42]. The arrival of rapid and inexpensive genome sequencing has shaken up these classical approaches by enabling large-scale epidemiological studies on bacterial virulence phenotypes[25]. These studies infer evolutionary scenarios leading to functional consequences, such as emergence of virulence, based on comparative phylogenomic data[19,43–47]. Such sequence data can also be used in GWAS if they are linked to virulence phenotypes[15,48]. The picture emerging from these studies is that extra-intestinal virulence in *E. coli* results from a multigenic process depending on the phylogenomic background, with a major role of the iron capture systems[6]. However, little is known about the relative contributions of the VAGs according to specific lineages.

In the present work, we explored further the emergence of a specific extra-intestinal virulent lineage within *E. coli* STc58/CC87[19,20]. Interestingly, it belongs to the B1 phylogroup which usually encompasses mostly commensal and intestinal pathogenic strains[6]. We showed, using a mouse model of sepsis together with a genetic association analysis, that the virulence phenotype among CC87 is less severe than in classical B2 ExPEC but strongly correlated to the presence of the HPI. In addition, we did not evidence at the *Escherichia* genus level any role of the ColV plasmid except by the fact that it is bearing the *aer* and *sit* operons. The latter are also frequently and even predominantly found on the chromosome as in the ExPEC B2, D and F archetypes which are highly virulent in mice[15].

There is however strong epidemiological evidence for a role of ColV plasmids in virulence in several *E. coli* lineages, such as the ST58 already discussed[19], the ST131 clade B[49] and the ST95[24]. It can be hypothesized that these plasmids are involved in aspects of extra-intestinal virulence that are not explored with our animal model, or in commensalism, gut colonization being the first step of extraintestinal infections. Indeed, ColV plasmid curing experiments using clinical isolates in chicken (intramuscularly) and mouse (intraperitoneally) sepsis models as well in gut colonization in human volunteers showed an important role of the plasmid in pathogenicity and faeces survival[50]. Alternatively, this plasmid's association with ExPEC virulence and colonization might be restricted to carrying iron capture systems in some sequence types, ColV plasmids behaving as dissemination vectors of the plasmid-borne genes in other strains and in some cases on the strain chromosome.

As iron capture systems are key elements in the mouse sepsis assay, we wanted to understand their respective role and the evolutionary forces acting on them. In addition to the HPI, *sit* and *aer* operons, we took in consideration *iro* gene cluster that is also present on the ColV plasmids and on the chromosome. Our data show a hierarchy among the VAGs with the HPI having the preeminent role but *aer*, *sit* and possibly *iro* (Fig. 3) genes potentializing the HPI effect (Fig. 4). This hierarchy could be the result of complementarity between different functions encoded by the different gene clusters. Aerobactin, salmochelin and yersiniabactin are hydroxamate, catecholate and mixed-type siderophore, respectively[37], with a wide range of $Fe^{3+}$

affinity[51], whereas Sit is an iron (Fe$^{2+}$) and manganese transporter[40]. The production of these siderophores is influenced differently by environmental factors as the pH and the carbon sources[52]. Specific additional functions have also been described. As examples, yersiniabactin competes for zinc with calprotectin[53], binds copper[54], and HPI encoded FyuA[55] and IroN[56] are involved in biofilm formation.

Interestingly, functional redundancy and preeminent role of specific iron acquisition systems have been observed for ExPEC in a urinary tract infection mouse model, allowing niche specificity within the urinary tract[37]. It has also been observed that salmochelin and aerobactin contribute more to virulence than heme in a chicken infection model[57] and have a cumulative effect[9,57]. Discrepancies between these studies and our own data could result from the fact that (i) the HPI was not considered, (ii) the results were based on single strain KOs and (iii) animal models other than mice were used. In addition, the importance of the tested systems has been shown to depend on the strain used[14,58]. This could be due to different associations of systems present in the strains. Such predominance of specific iron capture systems in a given niche was also observed in other species. Aerobactin but not yersiniabactin or salmochelin enables survival of hypervirulent *Klebsiella pneumoniae* in mouse systemic infection[59] and the *sit* operon is required for systemic stages of infection in *Salmonella* Typhimurium[60] and *Yersinia pestis*[61]. An important feature to take into account is also the associated metabolic cost of the siderophore production that has been reported as system and strain specific in *E. coli*[62]. Thus, hierarchy, niche specificity and cumulative effect of iron capture VAGs seem common themes in extra-intestinal virulence.

We provide strong evidence that specific combinations of VAGs are selected in phylogroups and STs. The plasmidic VAGs are evolutionary recent, highly mobile and may be selected in some contexts. Our data are in agreement with the fact that *iutA*, *iucC* and *irp2* are, among the commensal *E. coli* VAGs significantly increased in frequency over 30 years in France, and subjected to diversifying selection with a dN/dS (ratio of the number of nonsynonymous substitutions to the number of synonymous substitutions) ranging from 1.3 to 2.9[63]. A striking feature of our work is the ST (phylogroup) specific patterns of co-occurrences (Figs. 5 and S10). The ColV plasmid is present in ST95, 69 and 58, rare in the ST131 and totally absent in the ST73. Mirroring this pattern, the chromosomal VAGs are highly prevalent in ST73 and 131 but in the latter without *iro*, and present in ST95 but without *aer*. Similar lineage-specific co-occurrence patterns of iron related VAGs have been recently shown in *Klebsiella* spp. isolates[64]. Interestingly, these patterns are niche specific (human clinical versus porcine isolates) and resulted from multiple arrival events. Such a marked pattern (i.e. on/off) indicates very strong antagonism at the genome level (plasmid vs chromosome) and/or between the end products of the genes, corresponding to epistatic interactions.

The role of the strain genetic background in the acquisition/maintenance of acquired genes has been recently documented for the CRISPR-Cas systems acquired by horizontal gene transfer[65] by showing inhibition of non-homologous end-joining repair system by Csn2 Cas protein from which encoding genes never co-occurred[66]. It has been also suggested an incompatibility between the *pks* island producing the genotoxin colibactin and some antimicrobial resistance genes in the ST95[67]. In the same line, antivirulence genes, originally described in *Shigella*, have now been reported in other species as *Salmonella* and *Y. pestis*. These antivirulence genes, located on the recipient genome, must be inactivated for expression of full virulence of horizontally acquired elements[68]. Their functions and the molecular mechanisms at play are very diverse and none involve iron metabolism. Such reductive evolution is the hallmark of highly specific niche-adapted organisms, such as the intracellular pathogen and human-restricted *Shigella* spp which belong to the *E. coli* species[69]. However, our data seems to support that some ExPEC strains/lineages are undergoing niche

adaptation, trading overall versatility[6,26] for increased virulence in specific contexts.

We evidenced a high rate of gene inactivation in the *E. coli* iron capture systems as compared to core genome housekeeping genes (Fig. 7), as reported in bacteria living in communities within specific environments[32,33]. It can be hypothesized that, due to their metabolic costs[62], siderophores may be maladaptive in environments in which they are functionally redundant and therefore are prone to inactivation. Indeed, salmochelin, the most inactivated siderophore, is mostly plasmid encoded (Fig. S12) and thus prone to mutations[70], has a high metabolic cost[62] and is not involved in our mouse sepsis model. For the two other siderophore systems, yersiniabactin and aerobactin, the high inactivation of biosynthesis genes contrasting with the low inactivation of the receptor genes is in agreement with the cheater hypothesis[33]. When such inactivations occur in the HPI, they are associated with the presence of stable chromosomic *iro* and/or *sit* but not *aer* genes, reinforcing the HPI as a major system. Recently reported likelihood framework to assess correlated evolution[71] will be useful to decipher the type (obligate or preferential sequential order, synergy, incompatibility) and strength of interactions between the various events (gains, inactivation).

Our work has some limitations. First, our CC87 data can suffer from a lack of power due to the small number of CC87 mouse tested strains. Our analysis does not exclude minor roles of other loci such as the other ColV plasmid genes. However, due to ethical considerations, it is difficult to increase the number of mice used as we already obtained a strong signal. Second, although a well-accepted model, our mouse sepsis assay cannot mimic the invasion step of the infection as we inoculate the bacteria directly in the neck, which is a non-pathophysiological process. Various Gram-negative sepsis mouse models are available, but none can provide insights in all the steps of bacteremia pathogenesis[72]. Third, the urinary portal of entry is the most frequent in human bacteremia. In this context, ColV plasmids may play a significant role as they have been shown to enhance growth in human urine and colonization of the murine kidney[73].

Nevertheless, we bring strong evidence that a cumulative effect of iron capture systems is involved in *E. coli* extra-intestinal virulence with the HPI being the major player. The prevalence, the co-occurrences and the genomic location of these VAGs depend on the phylogenetic lineage, supporting the role of epistasis in the emergence of virulence. Deciphering the molecular mechanisms involved will be a real challenge for the future years and could have implications in the development of vaccines against ExPEC based on siderophore receptors[74–76] or siderophores themselves[77].

## Methods

### *E. coli* CC87 dataset

The collection is composed of strains belonging to the CC87 (Institut Pasteur scheme numbering) which gathered strains from both ST58 ($n = 139$) and its sister group ST155 ($n = 93$), all from phylogroup B1. The strains were sampled from humans ($n = 125$), domestic ($n = 66$) and wild ($n = 21$) animals and environment ($n = 20$) (Supplementary Data 1). Human strains corresponded to commensals ($n = 64$), ExPEC ($n = 60$) and InPEC ($n = 1$) whereas animal strains were all commensals (Fig. S3). The geographic origin was diverse (Europe $n = 91$, America $n = 59$, Australia $n = 50$, Africa $n = 30$, and Asia $n = 2$). Of note, 26 ST58 strains previously described by Reid et al.[19] were also included. All our strains were sequenced on Illumina platforms. Genomes were annotated with Prokka using standard parameters[78]. Bioprojects and accession numbers of the genome are detailed in Supplementary Data 1. Fasta and gff files are available on figshare[79,80].

### Genome typing, VAGs/ARGs screening and annotation

We determined MLST of each CC87 strain using mlst[81,82] based on both the Warwick University[83] and the Pasteur Institute schemes[84].

The O-type, H-type and *fimH* of the strains were determined with Abricate[85] with 90% identity and 90% coverage based on ecoh[86], serotypefinder[87] and fimTyper[88] databases. We searched for virulence associated genes (VAGs) and antibiotic resistance genes (ARGs) as previously described using Abricate with 90% identity and 90% coverage[23]. Our virulence database was composed of VirulenceFinder[89], VFDB[90] and specific genes from extra-intestinal *E. coli*. The VAGs were classified into 6 main families, namely invasin, protectin, toxin, adhesin, iron acquisition, and miscellaneous. The number of VAGs per strain was compared between the CC87 subgroups using Kruskal-Wallis test. We also searched for point mutations responsible for betalactam and fluoroquinolone resistance using pointFinder[91]. Based on ARGs and mutation presence/absence we predicted resistance phenotype to various antibiotics: ampicillin, piperacillin-tazobactam, cefotaxim/ceftriaxone, cefepim, carbapenems, fluoroquinolones, gentamicin, amikacin, sulfamides, trimethoprim, chloramphenicol, tetracyclines, colistin (Supplementary Data 3). The number of predicted resistant strains among each CC87 subgroup was compared with a Fisher exact test.

### ColV plasmid inference and plasmidic sequences prediction

To infer the presence of ColV plasmids we searched for VAGs from 6 gene sets (*cvaABC/cvi, iroBCDEN, icuABCD/iutA, etsABC, ompT/hlyF, sitABCD*) using Abricate with 90% identity and 95% coverage as proposed by Reid et al.[19]. Gene sequences were retrieved from the plasmid pAPEC-O2-ColV (RefSeq: NC_007675.1). Then, from these results, we inferred the presence of ColV plasmids on the basis of Liu's criteria[49], i.e. if at least one gene from 4 of the 6 gene sets was present. We also predicted chromosomal and plasmidic sequences using PlaScope[22], which is a targeted approach to classify contigs based on a database of chromosome and plasmid sequences of *E. coli* (database available on Zenodo[92]).

### Core-SNP phylogeny of the CC87

We built a core-SNP phylogeny by aligning the CC87 genomes to the reference strain IAI1 (ST$^{WU}$1128-ST$^{IP}$294, non-CC87, phylogroup B1) using Snippy 4.4.0[93] and we filtered recombination with gubbins v2.3.4[94] using standard parameters. The recombination free alignment was used to compute a phylogenetic tree with FastTree v2.1.11[95] with the generalised time reversible (GTR) + gamma substitution model. Phylogenetic tree was annotated with Itol[96].

### Mouse model of sepsis

We assessed the intrinsic virulence of 70 CC87 strains using a mouse model as previously described[11]. In this model, 10 female mice OF1 of 14−16 g (4 week-old) from Charles River (L'Arbresle, France) are inoculated with $10^8$ *E. coli* cells subcutaneously in the neck and monitored for six days. Two control strains (CFT073 and K-12) killing 10 and zero mice over 10, respectively, were systematically added in each experimental series. In all cases, we obtained similar results for the controls, mice being killed only by CFT073 at the same time after the inoculation within a range of 4 h. The number of mice killed per strain is available in Supplementary Data 1. Of note, 24 strains have been tested in a previous work[15] in the same way, while the other 46 were tested in the present study (Supplementary Data 1). In total this corresponds to a representative sample of each CC87 subgroup: ST$^{WU}$58-ST$^{IP}$186 (*n* = 2/7), ST$^{WU}$58-ST$^{IP}$87-A (6/12), ST$^{WU}$155-ST$^{IP}$21 (28/93), ST$^{WU}$58-ST$^{IP}$87-B (13/57), ST$^{WU}$58-ST$^{IP}$24 (21/63). Mouse survival curves were compared with the log-rank test between the CC87 subgroups and the control strains. To compare the number of mice killed per strain according to the presence of the HPI and to the presence of VAGs or combination of VAGs we classified strains as "killer" if they killed at least 9 mice over 10[11]. Then, we computed odds ratios for this status "killer" as a function of the presence of the HPI for each VAGs combination.

The experimental protocol (APAFIS#4948) was approved by the French Ministry of Research and by the Ethical Committee for Animal Experiments, CEEA-121, Comité d'éthique Paris-Nord. Housing conditions for the mice were in agreement with the French law, with dark/light cycle, and constant ambient temperature (21 °C +/− 2 °C) and humidity (50% +/− 10%). The mice were inoculated in a blind experiment by the zootechnician that ignores the status of the strain.

### Genome-wide association study to identify virulence determinants

We performed a GWAS based on the results of the mouse model of sepsis to identify genetic determinants associated with virulence in the CC87. We ran the analysis with Pyseer using the unitigs calculated from the CC87 genome assemblies (unitig-caller 1.2.1) or the gene presence/absence calculated with Roary 3.12.0[97] and the number of mice killed by each strain (continuous phenotype from 0 to 10). Unitigs are a compact representation of the pangenome in the form of extended DNA sequences of variable length[98]. The population structure was taken into account using the FastLMM linear mixed-model[99] and the patristic distances between each pair of strains extracted from the recombination-free phylogenetic tree previously computed. The *p*-value threshold for each analysis (i.e. unitigs and gene presence/absence) was corrected using Bonferroni method considering the number of unique variant patterns as the number of multiple tests. These corrections resulted in *p*-values of 6.09E-07 and 1.56E-05 for unitigs and genes, respectively. Finally, to identify their location, we mapped back the statistically significant unitigs using bwa 0.7.17[100] and bedtools 2.30.0[101] to 3 fully sequenced reference genomes of *E. coli*: CVM_N16EC0879 (ST$^{WU}$58-ST$^{IP}$24-like, phylogroup B1, GenBank: CP043744.1, https://www.ncbi.nlm.nih.gov/assembly/GCA_008386415.1/), IAI1 (ST$^{WU}$1128-ST$^{IP}$294, non-CC87 phylogroup B1, Refseq: NC_011741.1, https://www.ncbi.nlm.nih.gov/assembly/GCF_000026265.1/), S88 (ST$^{WU}$95-ST$^{IP}$1, phylogroup B2, Genbank: CU928161.2, https://www.ncbi.nlm.nih.gov/assembly/GCA_000026285.1). The coordinates of the significant unitigs and genes were used to draw physical maps of the region of interest from the reference genomes CVM_N16EC0879 and IAI1 using Clinker[102] and geom_segment from ggplot2[103]. Input files and raw results of the GWAS analysis are available on figshare[104–108].

### Role of VAGs from ColV plasmid in virulence at the species level

To assess the role of each VAG used to infer the presence of ColV plasmids, we retrieved the genomes of 370 strains representative of the diversity of the *Escherichia* genus as well as the results of the GWAS previously done on these strains[15]. First, we screened the genomes for ColV plasmid VAGs as described above and classify contigs of each genome using PlaScope. Then, we constructed a blastN database from nucleic sequences of all ColV plasmid VAGs found among the 370 genomes. We ran a blastN alignment using this database and the unitigs as query and considered only perfect matches (i.e. 100% identity over 100% of the unitig length). From this analysis we were able to associate each unitig and its *p*-value/beta-value to a given VAG. We also performed a blastN alignment of all sequences of a given VAG depending on its predicted location to compare VAGs sequences within and between different locations (i.e. plasmids and chromosomes) (Figs. S5, S6).

### Distribution and location of VAGs at the species level among fully sequenced genomes of *E. coli*

To analyse more thoroughly the distribution of VAGs used to infer ColV plasmids, we determined their prevalence among the 2302 genomes of *E. coli* (taxid 562) available on Refseq on September 19th 2022. Both fasta and genbank files were retrieved using ncbi-genome-download[109]. Accession numbers of assemblies and replicons (chromosomes and plasmids) are available in Supplementary Data 11. ColV plasmid VAGs were screened as described above using Abricate. Then, for each couple of VAG/location in a given genome, we searched for

other co-occurrent VAGs. Results are presented as heatmaps and colored according to the frequency of the co-occurrences and prevalence of VAG pairs. To test for association between chromosomal and/or plasmidic VAGs (*fyuA*, *iroN*, *iucA* and *sitA*) in a given ST/STc or phylogroup, we computed odds ratios from a contingency table constructed from the presence-absence data of each gene pair to be compared. When the results are significant, a $0 < OR < 1$ reflects a negative association between two VAGs and an $OR > 1$ a positive association. We also tested the association between the presence of HPI and another iron capture system at the whole species level using Pagel's model[29] from the R package phytools[110]. This method uses a likelihood ratio test to compare two models in which binary traits evolve independently or in a dependent manner along a phylogenetic tree. We considered a system as present in a strain when all the genes of the focal system were detected. The phylogenetic tree was generated from the core gene alignment of the 2302 genomes of RefSeq using Iqtree v1.6.12[111] under the GTR + F + I + G4 model.

In a second step, we determined the site of insertion for chromosomal VAGs. To this end, we performed a pangenome analysis of the 2302 genomes, using Ppanggolin v1.2.74[112] and the annotations from RefSeq as input. The gene families were computed by clustering protein sequences with 90% identity and 90% coverage. Then, we built the pangenome graph and partitioned the pangenomes. Finally, we used PanRGP[27] to determine the regions of genome plasticity (RGP), which can be considered as genomic islands in the cases of chromosomal insertions. The RGP containing *iro*, *aer* and *sit* operons were identified using the function "align". We extracted the sequences and annotations of these RGP as well as 3000 pb upstream and downstream to determine the sites of chromosomal insertion. The pangenome file from Ppanggolin/PanRGP is available on figshare[113].

Third, we also took advantage of this high-quality dataset to check for the performances of PlaScope. We analyzed the 2302 fully sequenced genomes with this approach and compared the replicon assignments to those of RefSeq (Fig S9).

## Phylogenic incongruence of VAGs among the 2302 genomes of RefSeq

We extracted the nucleic sequences of three operons involved in iron acquisition: *iroBCDEN*, *iucABCD* and *sitABCD*. We aligned the sequences of these operons according to their location (i.e. chromosomal or plasmidic) using Mafft v7.310[114] and computed phylogenetic trees with FastTree v2.1.11[95] with the GTR + gamma substitution model. From these trees, we computed patristic distances between all possible pairs of sequences according to their location and ST/phylogroup using "cophenetic" from the R package ape[115]. The trees were also annotated and visualized with Itols[96].

## Identification of inactivating mutations (nonsense, small and large deletion and insertion)

We retrieved the nucleotide and protein sequences of the four iron capture systems from the pangenome analysis (at the exception of the HPI integrase which matched with other integrases). We added the nucleotide sequences of incomplete protein that we detected with Abricate[85] with 50% identity and 50% coverage. In such few cases, we could retrieve the protein sequence of incorrectly annotated sequences [*iroC* ($n = 3$), *iroD* ($n = 3$), *iroE* ($n = 3$), *iroN* ($n = 3$), *iucA* ($n = 5$), *iucD* ($n = 475$), *iutA* ($n = 1$), *sitA* ($n = 1$), *sitB* ($n = 3$), *sitC* ($n = 12$) and *sitD* ($n = 1$)]. As a control we also extracted genes from both MLST schemes (Warwick University and Institut Pasteur), and we added the nucleotide sequences of genes detected with Abricate[85] with 25% identity and 5% coverage [*recA* ($n = 1$) and *uidA* ($n = 34$)]. We aligned the nucleotide sequences with Mafft v7.310[114] from the R package ips[116]. Next, to detect insertion and deletion events, we compared the aligned nucleotide sequences to the reference nucleotide sequences (i.e. a nucleotide sequence with a corresponding protein sequence). Finally,

to detect nonsense mutations (premature stop codon) we translated sequences aligned to the reference sequences.

We tested the association between the inactivation of genes in one system and the presence of another system using Pagel's model[29]. A system was considered inactivated for a strain if at least one gene was inactivated (nonsense, small and large deletion and insertion) and if no additional intact copy of the gene was found in the strain. For each system tested, we restricted the dataset to only strains carrying this system.

For each gene, we computed the inactivation rate as the number of strains with at least one inactivation event over the number of strains carrying the gene. This rate was also computed for the conserved MLST genes. Next, we computed binomial proportion confidence intervals (CI) for these rates. Finally, we obtained the corrected inactivation rate and CI by dividing the rate and CI by gene length. To evaluate the difference in corrected inactivation rate between gene categories (HPI, *iro*, *aer* and *sit* operon genes and MLST genes) and between function categories (biosynthesis, receptor, transport and other) we fitted linear models for the log-corrected inactivation rate (continuous variable) as a function of gene or function categories weighted by the inverse of the squared precision of the rate estimates. We used the log-transformed inactivation rate in the linear model to better respect the assumption of the linear model (normality of error). In this analysis, 0 rates were conservatively set at the upper bound of their confidence intervals.

### Reporting summary
Further information on research design is available in the Nature Portfolio Reporting Summary linked to this article.

### Data availability
The bioprojects and accession numbers of the CC87 genomes are available in Supplementary Data 1. The assemblies and annotation of the 370 strains of *Escherichia* are available on figshare (https://doi.org/10.6084/m9.figshare.11879340.v1, https://doi.org/10.6084/m9.figshare.19536163.v1). The 2302 complete genomes of *E. coli* can be obtained from RefSeq (https://www.ncbi.nlm.nih.gov/assembly) using the accession numbers available in Supplementary Data 11. Source data are provided with this paper.

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

## Acknowledgements

We are grateful to Laurence Armand and Milen Milenkov for the access to the Madagascar strains and to Lucile Vigué for fruitful discussions on selection and epistasis. This work was partially supported by the "Fondation pour la Recherche Médicale" Equipe FRM 2016, grant number DEQ20161136698 to E.D. M.G. was funded by the Deutsche Forschungsgemeinschaft (DFG, German Research Foundation) under Germany's Excellence Strategy - EXC 2155 - project number 390874280. G.R. was partially supported by the Agence Nationale de la Recherche under the French national action Plan on AntiMicrobial Resistance (PAMR) – project SEQ2DIAG (ANR-20-PAMR-0010).

## Author contributions

G.R., O.C. and E.D. concepted and realized the research. B.C. helped in the manipulations of the sequences. S.D. performed the mouse experiments. J.M. performed analysis on inactivating mutations and helped in the edition of the paper. F.B. performed the statistical analyses and edited the paper. M.G. was involved in the analysis of the GWAS data and the editing of the paper. G.R. and E.D. wrote the paper.

## Competing interests

The authors declare no competing interests.
