## [Peer Review File · Nature Communications]

Epistatic interactions between the high pathogenicity island and other iron uptake systems shape *Escherichia coli* extra-intestinal virulenceReviewer #1 (Remarks to the Author):

Royer et al. have investigated the genomes of STc 58 strains (a recently emerged complex of phylogroup B1 pathogenic strains). Genomes of STc 58 strains have also recently been investigated by Reid et al., who found that this group almost always carries ColV plasmids (which often encode siderophore genes) and the Yersiniabactin high-pathogenicity island. In this work, the authors examined a subset of the ST58 strains and identified five subgroups, one of which (ST58/24) possesses the majority of antibiotic resistance and virulence genes (which is consistent with Reid et al.). Importantly, they investigated virulence in mice of 70 of these strains (from all five groups). The ST58/24 group were more virulent in mice than the others but not as virulent as their B2 control strain. GWAS of this group to identify genes positively and negatively associated with virulence in the mouse infections clearly points to the HPI and not the ColV plasmids, which is a substantial step forward in understanding their role in bloodstream survival of these strains. However, as they note, their infection model cannot rule out a role for other genes on the ColV plasmid in natural translocation of ExPEC from the GI tract or other processes critical for these bacteria. They determined the location of the virulence-associated genes in a larger set of strains and found that in B1, they were almost always on plasmids but in the B2 and D strains they were mainly chromosomal. The most significant finding in this work was the relationship between single or combinations of iron-acquisition genes and virulence. Strains with the HPI (yersiniabactin) as well as *sitA*, *sitA/iucA*, or *sitA/iucA/iroN* tended to greater virulence than those same combinations but lacking HPI. Interestingly, the data suggest that acquisition of *sitA* in a strain without the HPI might serve as an anti-virulence factor, although this possibility was not considered explicitly. This could be tested directly as there were some strains that are fully virulent but lack any iron-acquisition system including HPI, which is contradictory to their claim that "HPI is necessary but not always sufficient for a strain to be virulent". The classic ExPEC STs appear to have multiple virulence associated genes in the chromosomes rather than plasmids. They provide convincing evidence that there is selection for acquisition of multiple virulence genes inserting at multiple sites, and that there is recent chromosomal gene sharing even among different lineages of these virulence factors.

Line 110: grouping by Reid et al was done by fastbaps whereas MLST groupings here were performed with BIGSdb and tseemann/mlst. It would be helpful for non-experts to know how these differ and whether this could impact the groupings.

Line 176 (Figure 3): It isn't readily apparent how this analysis (GWAS to find unitigs associated with virulence in mice) is different from Figure 2C, other than breaking each gene out individually and testing the effect of location.

Line 188: It seems as though the authors want to overstate the claims of Reid et al. with respect to the association of ColV plasmids with emergence of ST58c. As I read it, Reid et al. do not claim that ColV plasmids necessarily enable survival of the bacteria in blood or other extraintestinal sites but could increase fitness in the GI tract or other environments not tested in this mouse bloodstream infection model. (See also line 318-319).

Lines 198-202: Some of the sentences are confusing and should be reworded for clarity.

Line 229: Please add supporting references that ST10 strains are primarily commensals, particularly any experimental studies testing their ability to kill mice or cause disease in other model systems.

Line 256-257: This statement is very confusing. Does it mean that plasmids carrying *iro*, *iuc*, and *sit* do not arise in strains without the HPI? Could this be related to a possible anti-virulence effect of *sit* when HPI is not present?

Line 267: As written, this sentence implies that a single ST can belong to multiple phylogroups. Is this what is meant?

Lines 353-362: It would be useful for the authors to propose plausible potential antagonisms and the mechanisms involved that are observed in their work (among the iron acquisition systems) in addition to the example of incompatibility of non-homologous end joining and type II-A CRISPR-Cas systems which they cite.

Reviewer #2 (Remarks to the Author):

This is a fascinating manuscript that contains a lot of very good work. The analyses in most instances are robust and the marriage of phylogenetic data with animal models is commendable.

However animal models have their issues and their utility has been questioned when translating outcomes performed in animal studies to humans.

For example, in a Review entitled "The current state of animal models in research: A review" by Robinson et al., (<https://doi.org/10.1016/j.ijsu.2019.10.015>) the following statements were made "One review of over 60 highly cited animal studies in top journals between 1980 and 2000 found that only about a third translated to human randomized trials [27]. For example, there have been over 100 vaccines against HIV-like viruses developed which have demonstrated efficacy in animal models, however none, to date, have worked in humans [28–30]. In cancer research, the average rate of successful translation of animal research to human clinical trials is about 8% [31]"

Similarly, a recent review entitled "Pathogenesis of Gram-Negative Bacteremia" by Holmes et al., (*Clin Microbiol Rev* 2021 Mar 10;34(2):e00234-20. doi: 10.1128/CMR.00234-20.) shortcomings of animal models are highlighted in the section entitled "Modelling bacteremia".

Of note, genes carried on ColV plasmids seem to play a role in gastrointestinal colonisation. This is important as many of the ExPEC infections that result in humans are most likely to occur via faecal contamination of the urinary tract with potential ExPEC. Colonisation factors are important in this regard – this is sidestepped by the inoculation method used by Royer et al., which they acknowledge is a shortcoming. The importance of this model shortcoming is highlighted in work by Skyberg et al (<https://www.ncbi.nlm.nih.gov/pmc/articles/PMC1695531/>) demonstrating acquisition of ColV plasmids enhanced growth in human urine and colonisation of the murine kidney.

As stated above the animal model data is useful and is a strength in this manuscript but my point here is that one should be cautious in interpreting the outcomes of animal model studies.

So to summarise, while Royer et al., note some shortcomings in their manuscript these are not trivial given that the voracity of some of their statements is based on results using a single animal model. The authors perhaps should remember that ColV is a major virulence factor in Avian Pathogenic *E. coli* (APEC) that cause systemic infections (colibacillosis) that are major afflictions in the poultry industry. The linkage of ColV with human disease is also noted especially in lineages of ST131 Clade B (Liu et al., 2018 *mBio*. 2018 Aug 28;9(4):e00470-18. Doi: 10.1128/mBio.00470-18) and in lineages in ST95 (Cummins et al., 2022). Finally, we wonder how the data interpretation may be influenced by choice of animal model, noting that there is a reliable avian model for ExPEC disease, which was also noted by the authors.

Nonetheless, we have no issue with the analyses but finds aspects of the interpretation of the data somewhat confusing. As I read this manuscript, I find that it is in near complete concordance with the work performed by Reid et al., - a paper that the authors go to great lengths to compare. However, they seem to take issue with the important role of ColV plasmids as carriers of critically important VAGs that together with HPI facilitate ExPEC disease in humans and in poultry (and possibly in swine) despite coming to the same conclusion in their manuscript. Moreover, there is insufficient data to categorically discount a role for ColV plasmids in the evolution of *E. coli* virulence especially in emerging lineages that belong to non B2/D phylogroups.

Overall, we feel this sentiment stems from the following:

1. Some minor misconceptions about our paper:
 - a. That we view ColV plasmids themselves to be a "main driver of virulence" as opposed to an important carrier of VAGs in certain *E. coli* lineages
 - b. That we rank ColV and their iron acquisition-related genes above the HPI in terms of importance regarding evolution and/or virulence

2. The STc58 GWAS analysis not showing any association between ColV presence or ColV genes (incl. sit/aer) being significantly associated with virulence (Line 151)
3. The genus level GWAS analysis finding an association between sit/aer that was agnostic of ColV presence/other ColV-carried genes

Here, we will address the above points in more detail by highlighting extracts from the manuscript:

Line 81- "Using phylogenomic comparative analysis, Reid et al. point to a heightened role of ColV plasmid over the HPI in the evolution of this clone."

- Whilst our manuscript was obviously focused on ColV plasmids, we did not rank any virulence factor or mobile element over another in terms of importance, and also highlighted the presence and potential importance of the HPI. "Carriage of yersiniabactin (fyuA, irp2, HPI) by the vast majority of strains in the BAP2 cluster also strongly supports their innate virulence". Royer et al., somewhat contradict this statement later in the manuscript – see below.

Regarding the GWAS analysis of ST58:

Line 151 "In contrast, no unitig or gene related to the ColV plasmids were significantly associated with virulence (Figure 2C)."

- We note that in this analysis more than half of the strains were not actually ST58, which makes the results somewhat incomparable to ours regarding the importance of ColV and the sit/aer genes they carry because we only analysed ST58. Is it not possible that the inclusion of mostly non-ST58 strains could explain a lack of association between sit/aer and virulence that is present within ST58 sensu stricto? Perhaps the authors could perform the GWAS for the 34 ST58 strains and see if the result holds?

Royer et al., go on to use GWAS to indicate the importance of plasmidic factors- especially sitA-D and iucA-D/iutA both of which are on ColV plasmids in the following statement:

Lines 177-187 "Then, using GWAS, we looked at associations of unitigs belonging to each of these VAGs with virulence in mice. Interestingly, we found significant associations mainly for the aerobactin and sit operons (Figure 3).If we consider the presence of the ColV plasmid as a covariate in the GWAS analysis, only the unitigs associated with iucABCD/iutA and sitABCD remain significant, arguing for a role beyond their simple localization on the plasmid (Figure S6)."

- This result is readily explained by the fact that a) these genes are mostly chromosomal in the B2/D dominated genus collection, and b) despite conserved virulence gene carriage, the remaining content of ColV plasmids can be highly heterogenous; including variable or absent replication/transfer regions and other accessory content such as AMR that would not be expected to associate with virulence. It does not negate the importance of the ColV backbones in carrying and transmitting the conserved virulence associated genes within and between strains that do not possess them chromosomally. Whether these genes can also have a chromosomal location especially in B2 /D phylogroups is not relevant to the importance of ColV plasmids as mobile vehicles of important VAGs in other phylogroups and STs such as ST58. In addition, gene dosage effects are not considered in the argument (carriage of iucABCD/iutA and sitABCD on plasmid and chromosome)- which conceivably may be significant.

Royer et al., conclude:

Lines 188-192. "Overall, these data indicate that within the Escherichia genus, the sole presence of the ColV plasmids has little or no role in virulence in the mouse model, with the exception of the two aerobactin and sit operons. However, the predominant location of the latter is in fact chromosomal, reinforcing the idea that ColV plasmids are not themselves the main driver of virulence.

- Many emerging pathogens come from commensal or non-B2/D phylogroups, and we argue that ColV carriage is likely to be important in this evolutionary trajectory due to the carriage of

important virulence genes identified in this analysis. These lineages are unlikely to have extensive chromosomal pathogenicity Islands (otherwise they would NOT be commensal phylogroups). Here, we also sense the misconception from our paper that it was asserted that 'ColV plasmids are the main driver of virulence', which we do not believe, as stated above they are clearly important because they carry and transmit these genes in strains that don't carry them on the chromosome.

Royer et al., provide further critical data that lends weight to the importance of ColV plasmids as vectors of VAGs in the section The HPI needs other VAGs to express full virulence

Line 204 "The avirulent strains despite the presence of the HPI belonged to phylogroups A (n=5/9), B2 203 (n=2/9), C (n=2/9) (Figure S7). Moreover, when we compared strains carrying the HPI depending on the presence of the other operons, the presence of the HPI alone was associated with significantly fewer mice killed per strain than in the cases of the HPI with sit, sit/iro, sit/iuc or sit/iuc/iro. No differences were observed for strains carrying only iro, iuc or iro/iuc probably due to the limited number of strains.

Overall, these data indicate that the presence of HPI is necessary but not always sufficient for a strain to be virulent, with other iron uptake systems required to express full virulence. Conversely, these accessory systems alone are not sufficient to acquire full virulence.

Finally, the authors highlight the importance of ColV in ST58 in the following section:

Lines 228-239. "Moreover, these plasmidic VAGs usually occurred in strains which carry *fyuA*, except in the commensalism-associated ST10. More globally, we can see specific patterns. In the case of ST131, the overall pattern is dominated by a very high prevalence and frequent co-occurrence of chromosomal VAGs, with a notable absence of *iroN*. This chromosomal dominant pattern is exacerbated in the ST73 with an almost complete fixation of VAGs on the chromosome. The ST69 and ST95 both present high prevalence and co-occurrence frequencies of chromosomal and plasmidic VAGs, without chromosomal *iroN* or *iucA*, respectively. Of note, this is in agreement with the ColV plasmids frequently observed in the ST95. On the opposite and consistent with its commensal behaviour, the ST10 carries few VAGs, which are usually clustered on plasmids. Finally, for the STc58, in accordance with our previous observation and data from Reid et al., *iroN*, *iucA* and *sitA* co-occurred mainly and frequently on plasmids and are usually associated with the HPI."

Royer et al., clearly stated that our paper highlighted the co-occurrence of HPI and ColV- which carry *iroN*, *iucA* and *sitA* which is at odds with their concluding statement in the Introduction ('Using phylogenomic comparative analysis, Reid et al. point to a heightened role of ColV plasmid over the HPI in the evolution of this clone')

In conclusion, Royer et al., provide evidence that supports a major conclusion from our paper that is encapsulated in the title "A role for ColV plasmids in the evolution of pathogenic *Escherichia coli* ST58" given the lack of chromosomal virulence factors in commensal *E. coli* phylogroup lineages that have emerged to cause serious disease in humans.

For clarity Royer et al., may consider highlighting the disparities in the selection of genomes used for their GWAS analysis. It is quite difficult to keep track of which results sections/data were generated on which collection – there are four under analysis:

1. CC87
2. 370 phylogenetically diverse strains
3. 2200 genomes from refseq
4. A collection (n=?) of ST131, ST73, ST69, ST10, ST95 and STc58

It should be noted that the collection of 370 genomes is 30% B2, 19% A, 11% B1, 10% C and 5% D etc. The collection of ~2200 is 33% A, 21% B1, 15% B2, 13% E and 9% D etc. ColV is likely to play a key role in the emergence of pathogenic commensal phylogroups that typically lack chromosomal virulence islands but that have acquired HPI so the percentage of B2 in any analysis will cloud interpretation of the importance of plasmidic VAGs.

The point of all this is that the authors might consider revising and nuancing some of their comments about ColV, as their data and ours show that they carry important VAGs that play a role in supplementing the role of HPI as a key virulence island.

Reviewer #3 (Remarks to the Author):

Royer and colleagues present a genomic analysis of the evolution of virulence associated genes in *E. coli*, with a special focus on iron acquisition systems. Key to many of the findings was a previously generated compendium of mouse experiments evaluating the outcome of bloodstream infections with ~300 diverse *E. coli* strains. The authors begin by focusing on the emergent ST58 B1 lineage and perform a genome-wide association study to identify genes/variants associated with death of mice in the aforementioned experiments. The only hit identified was the previously described high pathogenicity island (HPI), harboring the yersiniabactin iron scavenging system. The authors made especial note that they did not identify the colV plasmid to be associated with virulence, contrasting/conflicting with a recent report in *Nature Communications* (Reid et al., 2022). The authors then go on to examine the role of virulence associated genes, and col plasmids, more broadly in *E. coli* virulence by performing GWAS in the diverse *E. coli* lineages from the compendium. These GWAS analyses showed a strong signal for the presence of HPI, as well as other iron scavenging systems (*iuc* and *sit*), in a manner that appears dependent on the presence of HPI. The authors further show that these GWAS hits are independent of whether the associated genes are encoded on the chromosome or plasmids. Lastly, the authors examine the co-association of different virulence genes, and show that they were all acquired multiple times independently and display different patterns of co-association in different *E. coli* lineages. The authors use this observation to infer that there is a strong selective pressure to acquire these genes, and that there are likely genetic interactions among them that influence their patterns of association.

Overall, this is an interesting analysis, marrying comparative genomics and GWAS to test specific hypotheses regarding the relative roles of different operons in a virulence proxy. However, it is important to note that several of the key observations are more confirmatory of previous publications; association of iron scavenging loci with death in this mouse compendia were previously reported by the authors (Galardini et al., *PLoS Genetics*, 2020) and the observation of HPI being a lineage defining trait for virulent ST58 sub-lineage was reported by Reid et al. in 2022. The two key novel findings are potential interactions among the iron scavenging loci in this mouse model of bloodstream infection, and the rebuttal of the role of colV plasmids in the emergence of pathogenic ST58. The first observation is potentially interesting (although limited to virulence in the context of bloodstream infections) and is a clever use of the existing data set. With regards to the second observation, while I agree that the Reid et al. paper is limited by having no experiments (i.e. just enrichment of genetic loci among exPEc isolates), the current study has its own significant limitations in only focusing on the fitness in the bloodstream (i.e. disregarding the potential role of colV in earlier stages of infection).

Major comments

1. There are a couple of instances where the authors make statements based on Figures, but I did not see statistical support for the assertions. Three in particular were: i) the survival curves in figure 2, ii) difference in virulence for strains having just HPI versus HPI + another virulence genes in Figure 4 and ii) *iro/iuc/aiu* appearing in the presence of HPI, but not vice versa, in Figure 5. For the first and third point please provide statistical support, and for the second please quantify.

2. In the discussion the authors conjecture regarding the implications of gene co-occurrence/exclusion in Figure 6 on potential genetic interactions. Given that the interactions described are primarily among iron scavenging loci, I think the discussion would benefit from

expanding on how the varying functions/affinities of these systems may account for their distributions, versus unspecified detrimental interactions that may result in their association/exclusion.

3. The visualization of the nucleotide diversity in Figure 6 is quite interesting and indicates that the copies of iron scavenging genes on plasmids trace back to a single mobilization event. It would be interesting to see where the corresponding plasmid associated operons fit onto the chromosomal tree, to gain insight into their origin.

Reviewer comments and author answers (in red)

Reviewer #1 (Remarks to the Author):

However, as they note, their infection model cannot rule out a role for other genes on the ColV plasmid in natural translocation of ExPEC from the GI tract or other processes critical for these bacteria.

This point is now largely discussed (see below).

The most significant finding in this work was the relationship between single or combinations of iron-acquisition genes and virulence. Strains with the HPI (yersiniabactin) as well as *sitA*, *sitA/iucA*, or *sitA/iucA/iroN* tended to greater virulence than those same combinations but lacking HPI. Interestingly, the data suggest that acquisition of *sitA* in a strain without the HPI might serve as an anti-virulence factor, although this possibility was not considered explicitly. This could be tested directly as there were some strains that are fully virulent but lack any iron-acquisition system including HPI, which is contradictory to their claim that “HPI is necessary but not always sufficient for a strain to be virulent”.

The concept of antivirulence is very interesting and now discussed in the discussion section of the paper with several arguments against this hypothesis. We do not think that with the small numbers of strains tested in mice we can demonstrate an antivirulence role for *SitA* (see below). Nevertheless, we looked for the inactivation of the genes of the four systems and found an increase of gene inactivation as compared to housekeeping genes. We also found a specific association of the inactivation of the HPI with the presence of several iron systems. All these data are now presented and discussed accordingly (see below). It can be hypothesized that, due to their metabolic costs, siderophores may be maladaptive in environments in which they are functionally redundant and inactivated.

For the contradictory claim, we changed the sentence for “...in most cases, the presence of the HPI is necessary but not always sufficient...”

Line 110: grouping by Reid et al was done by fastbaps whereas MLST groupings here were performed with BIGSdb and tseemann/mlst. It would be helpful for non-experts to know how these differ and whether this could impact the groupings.

Multi-locus sequence typing (MLST) and fastbaps analyses both aim at clustering microbial population from nucleic sequences. MLST is based on a limited number of partial gene sequences ($n=7$ or 8 for *E. coli*) which are converted into numbered alleles which are then combined into a profile. This profile is expressed by a number which corresponds to the Sequence Type (ST) of the strain studied as defined by an international nomenclature. The advantages of MLST are that it is portable, highly reproducible with an international nomenclature (pubmlst.org). It is not affected by the input sequences (sampling, phylogeny). Furthermore, because of the conversion of sequences into alleles, it is less affected by the phenomenon of recombination. For all these reasons, it is frequently used to characterize and compare globally distributed lineages such as *E. coli* ST131 or *K. pneumoniae* ST258. Fastbaps combines a Bayesian Hierarchical Clustering (BHC) and the hierBAPS algorithm to perform model-based clustering from a core genome alignment. It has the advantage of being very fast and robust when clustering large datasets and of including different clustering models. However, it is less portable than MLST because the cluster definition is depending of the strain sampling, their phylogeny and the consideration of recombination. In our work, the classification is based both on the phylogenetic history of the strains with an external root and the classical MLST using the Warwick University and Institut Pasteur schemes.

We now provide a Figure S1 with two phylogenetic trees to highlight differences in fastBAPs clustering according to the alignment (core gene alignment without filtering for recombination vs recombination-free coregenome alignment). As can be seen from these trees, the clusterings differ both in terms of composition

and number of clusters (outermost circle). Moreover, these clusters also differ from the BAP group defined by Reid *et al.* (genomes with colored label background). This latter observation probably also results from differences in the strain sampling. Nevertheless, our phylogeny/MLST approach and the fastBAPs approach of Reid *et al.* are in agreement in defining the BAP2 cluster described by Reid *et al.* as a well-supported phylogenetic lineage that carry many virulence and resistance associated genes.

We now modified the main text as follow and added a supplementary figure showing the discrepancies:

“The small discrepancies between Reid *et al.* clustering and our subgroup definition are probably due to differences in (i) the clustering approaches (fastbaps model-based clustering vs combination of phylogeny and multi-locus sequence typing), ii) the alignment used to compute the phylogenetic tree (core gene alignment without filtering for recombination vs recombination-free core genome alignment) (Figure S1) and (iii) the strain sampling (ST58 sensu stricto versus CC87). Importantly, the ST58/24 encompasses the BAP2 cluster described in 18.”

Line 176 (Figure 3): It isn't readily apparent how this analysis (GWAS to find unitigs associated with virulence in mice) is different from Figure 2C, other than breaking each gene out individually and testing the effect of location.

The Figure 3 presents the results of the GWAS performed on the whole species (n=370 strains) considering both the phylogroups and each gene individually. The Figure 2 focused on a different dataset composed of 232 B1 phylogroup CC87 strains. We have now modified the legend of the Figure 2 as follows to make it clearer:

“From genotypic and phenotypic characterization of CC87 extra-intestinal virulence to identification of genetic determinants by GWAS in the 232 B1 phylogroup CC87 strains.”

Line 188: It seems as though the authors want to overstate the claims of Reid *et al.* with respect to the association of ColV plasmids with emergence of ST58c. As I read it, Reid *et al.* do not claim that ColV plasmids necessarily enable survival of the bacteria in blood or other extraintestinal sites but could increase fitness in the GI tract or other environments not tested in this mouse bloodstream infection model. (See also line 318-319).

Line 188, we have now insisted that what we observed is specific to our model and removed the sentence lines 190-192.

We have also removed the sentence lines 318-319 and discussed in a more balanced way the potential role of the plasmid, including a role as vector of dissemination and in other environments as urine and the GI tracts in the Discussion section.

Lines 198-202: Some of the sentences are confusing and should be reworded for clarity.

The text has been modified taking into account both reviewer 1 and reviewer 3 comments. The results have been presented in a clearer way. The Figure 4 and Figure S8 have also been modified to add statistical support and to clarify the association of *iroN*, *iucA* and *sitA* (see below).

Line 229: Please add supporting references that ST10 strains are primarily commensals, particularly any experimental studies testing their ability to kill mice or cause disease in other model systems.

We removed this sentence. A similar sentence 7 lines below is now referenced with experimental and epidemiological data supporting the commensal character of ST10 in human (PMID: 30085181, PMID: 20157339).

Line 256-257: This statement is very confusing.

We agree that the statement was confusing. We modified the text.

Does it mean that plasmids carrying *iro*, *iuc*, and *sit* do not arise in strains without the HPI?

The term “arise” was unsuitable as we do not have robust evidence of the temporality of the events. We have now changed the sentence to “Interestingly, in non-typical ExPEC (STc58, A, B1, C, E, G) plasmidic *iro*, *iuc* and/or *sit* operons are positively associated with *fyuA*.”

Could this be related to a possible anti-virulence effect of *sit* when HPI is not present?

We greatly acknowledge the reviewer for this hypothesis. We agree that according to the figures 4 and S8, no mouse is killed when *sitA* (alone or in association with *iroN* and/or *iucA*) is present without the HPI. On the contrary, mice are massively killed by strains carrying the HPI. However, such effect is also observed in strains that do not carry any VAG (*iroN*, *iucA* and *sitA*) (upper left plot from figures 4 and S8), arguing against this hypothesis and favoring a major role of the HPI in virulence. Nevertheless, we now discuss this point of anti-virulence and searched for inactivation of the iron capture systems studied (see above and below).

Line 267: As written, this sentence implies that a single ST can belong to multiple phylogroups. Is this what is meant?

Indeed, this sentence was confusing. We modified it as follow:

“These sites were mainly ST specific among a given phylogroup. As an example, in strains from ST127 (B2 phylogroup) *iro* gene cluster is inserted into the tRNA-SerW whereas in other B2 strains the insertion site is almost exclusively in the tRNA-SerX (Table S5). However, STs belonging to different phylogroups sometimes shared the same site.”

Lines 353-362: It would be useful for the authors to propose plausible potential antagonisms and the mechanisms involved that are observed in their work (among the iron acquisition systems) in addition to the example of incompatibility of non-homologous end joining and type II-A CRISPR-Cas systems which they cite.

Unfortunately, we don't have solid hypothesis to propose potential antagonism between iron capture system. We have now extended this section describing the species, their life styles and the mechanisms involved in antivirulence and gave arguments against this hypothesis. However, we now have explored the presence of inactivating mutations in the HPI, *aer*, *iro* and *sit* genes. Overall, the inactivation rates of these genes differ from the MLST housekeeping gene ones (with the exception of *uidA* which is known to be inactivated in O157:H7 enterohemorrhagic strains). We also identified a significant association between inactivation in HPI and the *iro* and/or *sit* chromosomal gene clusters. We now discuss of a possible hypothesis for these inactivations based on the metabolic costs of the systems.

Reviewer #2 (Remarks to the Author):

This is a fascinating manuscript that contains a lot of very good work. The analyses in most instances are robust and the marriage of phylogenetic data with animal models is commendable.

We would to thank the reviewer #2 for her/his kind comments.

However animal models have their issues and their utility has been questioned when translating outcomes performed in animal studies to humans.

For example, in a Review entitled “The current state of animal models in research: A review” by Robinson et al., (<https://doi.org/10.1016/j.ijsu.2019.10.015>) the following statements were made “One review of over 60 highly cited animal studies in top journals between 1980 and 2000 found that only about a third translated to

human randomized trials [27]. For example, there have been over 100 vaccines against HIV-like viruses developed which have demonstrated efficacy in animal models, however none, to date, have worked in humans [28–30]. In cancer research, the average rate of successful translation of animal research to human clinical trials is about 8% [31].

Similarly, a recent review entitled “Pathogenesis of Gram-Negative Bacteremia” by Holmes et al., (Clin Microbiol Rev 2021 Mar 10;34(2):e00234-20. doi: 10.1128/CMR.00234-20.) shortcomings of animal models are highlighted in the section entitled “Modelling bacteremia”.

We totally agree with these comments. Animal models are far from the real life in human.

Of note, genes carried on ColV plasmids seem to play a role in gastrointestinal colonisation. This is important as many of the ExPEC infections that result in humans are most likely to occur via faecal contamination of the urinary tract with potential ExPEC. Colonisation factors are important in this regard – this is sidestepped by the inoculation method used by Royer et al., which they acknowledge is a shortcoming. The importance of this model shortcoming is highlighted in work by Skyberg et al (<https://www.ncbi.nlm.nih.gov/pmc/articles/PMC1695531/>) demonstrating acquisition of ColV plasmids enhanced growth in human urine and colonisation of the murine kidney.

We now had added caution to our results by citing the review by Holmes *et al.*, and discussing the potential role of ColV plasmids in urinary tract infection and gastro-intestinal colonization.

As stated above the animal model data is useful and is a strength in this manuscript but my point here is that one should be cautious in interpreting the outcomes of animal model studies.

We totally agree with this and hope that the manuscript is now more balanced.

So to summarise, while Royer et al., note some shortcomings in their manuscript these are not trivial given that the voracity of some of their statements is based on results using a single animal model.

We sincerely apologize to have given an impression of negativity about the work done by Reid *et al.* Their work and ours are in fact complementary and consistent with each other.

The authors perhaps should remember that ColV is a major virulence factor in Avian Pathogenic E. coli (APEC) that cause systemic infections (colibacillosis) that are major afflictions in the poultry industry.

The role of ColV plasmids is now discussed using the recently published work of Mageiros *et al.* (PMID: 33536414).

The linkage of ColV with human disease is also noted especially in lineages of ST131 Clade B (Liu et al., 2018 mBio. 2018 Aug 28;9(4):e00470-18. Doi: 10.1128/mBio.00470-18) and in lineages in ST95 (Cummins et al., 2022).

We have now cited these two papers as providers of epidemiological evidences for a role of ColV plasmids in virulence.

Finally, we wonder how the data interpretation may be influenced by choice of animal model, noting that there is a reliable avian model for ExPEC disease, which was also noted by the authors.

This point is now discussed in the conclusion section.

Nonetheless, we have no issue with the analyses but finds aspects of the interpretation of the data somewhat confusing. As I read this manuscript, I find that it is in near complete concordance with the work performed by Reid et al., - a paper that the authors go to great lengths to compare. However, they seem to take issue with

the important role of ColV plasmids as carriers of critically important VAGs that together with HPI facilitate ExPEC disease in humans and in poultry (and possibly in swine) despite coming to the same conclusion in their manuscript. Moreover, there is insufficient data to categorically discount a role for ColV plasmids in the evolution of *E. coli* virulence especially in emerging lineages that belong to non B2/D phylogroups.

Overall, we feel this sentiment stems from the following:

1. Some minor misconceptions about our paper:

a. That we view ColV plasmids themselves to be a “main driver of virulence” as opposed to an important carrier of VAGs in certain *E. coli* lineages

b. That we rank ColV and their iron acquisition-related genes above the HPI in terms of importance regarding evolution and/or virulence

2. The STc58 GWAS analysis not showing any association between ColV presence or ColV genes (incl. *sit/aer*) being significantly associated with virulence (Line 151)

3. The genus level GWAS analysis finding an association between *sit/aer* that was agnostic of ColV presence/other ColV-carried genes

Overall, we are now presenting our data and discussing the work of Reid *et al.* in a more balanced way. Indeed, we put several notes of caution in the interpretation of our results and present now the two studies as complementary. More precisely, we mitigated our conclusion in the third paragraph of the discussion and in the limitation paragraph.

Here, we will address the above points in more detail by highlighting extracts from the manuscript:

Line 81- “Using phylogenomic comparative analysis, Reid *et al.* point to a heightened role of ColV plasmid over the HPI in the evolution of this clone.”

- Whilst our manuscript was obviously focused on ColV plasmids, we did not rank any virulence factor or mobile element over another in terms of importance, and also highlighted the presence and potential importance of the HPI. “Carriage of yersiniabactin (*fyuA*, *irp2*, HPI) by the vast majority of strains in the BAP2 cluster also strongly supports their innate virulence”. Royer *et al.*, somewhat contradict this statement later in the manuscript – see below.

We have now changed this sentence as follow:

“Using phylogenomic comparative analysis, Reid *et al.* point to a role of ColV plasmid, together with the HPI, in the evolution of the pathogenicity of this clone.”

Regarding the GWAS analysis of ST58:

Line 151 “In contrast, no unitig or gene related to the ColV plasmids were significantly associated with virulence (Figure 2C).”

- We note that in this analysis more than half of the strains were not actually ST58, which makes the results somewhat incomparable to ours regarding the importance of ColV and the *sit/aer* genes they carry because we only analysed ST58. Is it not possible that the inclusion of mostly non-ST58 strains could explain a lack of association between *sit/aer* and virulence that is present within ST58 *sensu stricto*? Perhaps the authors could perform the GWAS for the 34 ST58 strains and see if the result holds?

Indeed, in our study, we considered the whole STc58 to take into account the population structure of this clonal complex in the GWAS analysis. Such an approach is recommended to limit the identification of associations that rely solely on a lineage effect. Nonetheless, as requested by the reviewer, we performed the GWAS analysis including only the ST58 (n=34) while considering or not the population structure. Results are expressed with the p-value of the association, adjusted for population structure (lrt-pvalue from pyseer) and the beta-value which is the effect size of the unitigs/genes. When correcting for population structure, we did not find significant results either by the unitig-based approach (Figure 1A) or by the gene presence/absence approach (Table 1).

When we did not take population structure into account, we found 25 significant unitigs, among which 3 and 22 were positively and negatively associated with the phenotype, respectively (Figure 1B). Of the three positively associated unitigs, two match with the HPI and one match with the region just upstream of the integration. All the 22 negatively associated unitigs match with the insertion region of the HPI in strains that lack the yersiniabactin. Through the gene presence/absence approach, we found 17 significant genes, among which 13 and 4 were positively and negatively associated with the phenotype, respectively. All positively associated determinants were linked to the HPI while 3 of the 4 negatively associated determinants are in the insertion region of the HPI (*yodB*, *yeeJ*, tRNA-Asn) (Table 1). The remaining negatively associated gene encodes a protein of the DUF977 family, distant from HPI. We did not find a significant association for VAGs from the ColV plasmids.

Figure 1: Results of the unitig association with virulence when considering only strains belonging to ST58 *sensu stricto* A) with or B) without correction for population structure. The p-value of the association is shown on the y-axis, the effect size (beta) on the x-axis and the significance level with a dotted line. The unitigs found in genes belonging to the ColV plasmid or to the HPI are highlighted in green and red, respectively. Other unitigs are in grey.

Table 1: Results of the association of genes from the ColV plasmid and the HPI with virulence when considering only strains belonging to ST58 *sensu stricto* while correcting or not for population structure.

	Correction for population structure*		No correction for population structure**	
Genes	p-value	Beta-value	p-value	Beta-value

iroB	0.523	1.05	0.252	1.53
iroC	0.523	1.05	0.252	1.53
iroD	0.523	1.05	0.252	1.53
iroE	0.523	1.05	0.252	1.53
iroN	0.523	1.05	0.252	1.53
cvaA	0.664	0.698	0.354	1.25
cvaB	0.585	0.871	0.373	1.21
cvaC	0.523	1.05	0.252	1.53
etsC	0.523	1.05	0.252	1.53
etsB	0.523	1.05	0.252	1.53
etsA	0.523	1.05	0.252	1.53
ompT	0.523	1.05	0.252	1.53
hlyF	0.523	1.05	0.252	1.53
sitA	0.198	1.99	0.12	2.06
sitB	0.198	1.99	0.12	2.06
sitC	0.0491	2.77	0.0477	2.6
sitD	0.23	1.99	0.0973	2.21
iucA	0.0496	3.03	0.0273	2.91
iucB	0.293	1.84	0.144	1.94
iucC	0.0496	3.03	0.0273	2.91
iucD	0.0496	3.03	0.0273	2.91
iutA	0.23	1.99	0.0973	2.21
fyuA	0.000165	5.8	1.96E-05	5.17
irp5	0.000165	5.8	1.96E-05	5.17
irp4	0.000165	5.8	1.96E-05	5.17
irp3	0.000165	5.8	1.96E-05	5.17
irp1	0.000165	5.8	1.96E-05	5.17
irp2	0.000165	5.8	1.96E-05	5.17
ybtA	0.000165	5.8	1.96E-05	5.17
irp6	0.000165	5.8	1.96E-05	5.17
irp7	0.000165	5.8	1.96E-05	5.17
irp8	0.000165	5.8	1.96E-05	5.17
irp9	0.000165	5.8	1.96E-05	5.17
int	0.000165	5.8	1.96E-05	5.17
*p-value threshold after Bonferroni multiple testing correction: 3.19E-05				
**p-value threshold after Bonferroni multiple testing correction: 3.13E-05				

In conclusion, whatever the sampling and the consideration of population structure we do not see any role of ColV plasmid genes. However, this could be due to the very small number of strains (n=34) which could reduce the statistical power of the analysis.

Royer et al., go on to use GWAS to indicate the importance of plasmidic factors- especially *sitA-D* and *iucA-D/iutA* both of which are on ColV plasmids in the following statement:

Lines 177-187 “Then, using GWAS, we looked at associations of unitigs belonging to each of these VAGs with virulence in mice. Interestingly, we found significant associations mainly for the aerobactin and *sit* operons (Figure 3).If we consider the presence of the ColV plasmid as a covariate in the GWAS analysis, only the unitigs associated with *iucABCD/iutA* and *sitABCD* remain significant, arguing for a role beyond their simple localization on the plasmid (Figure S6).”

- This result is readily explained by the fact that a) these genes are mostly chromosomal in the B2/D dominated genus collection, and b) despite conserved virulence gene carriage, the remaining content of ColV plasmids can be highly heterogenous; including variable or absent replication/transfer regions and other accessory content such as AMR that would not be expected to associate with virulence. It does not negate the importance of the ColV backbones in carrying and transmitting the conserved virulence associated genes within and between strains that do not possess them chromosomally. Whether these genes can also have a chromosomal location especially in B2 /D phylogroups is not relevant to the importance of ColV plasmids as mobile vehicles of important VAGs in other phylogroups and STs such as ST58.

This point is now discussed.

In addition, gene dosage effects are not considered in the argument (carriage of *iucABCD/iutA* and *sitABCD* on plasmid and chromosome)- which conceivably may be significant.

Royer et al., conclude:

Lines 188-192. “Overall, these data indicate that within the *Escherichia* genus, the sole presence of the ColV plasmids has little or no role in virulence in the mouse model, with the exception of the two aerobactin and *sit* operons. However, the predominant location of the latter is in fact chromosomal, reinforcing the idea that ColV plasmids are not themselves the main driver of virulence.

- Many emerging pathogens come from commensal or non-B2/D phylogroups, and we argue that ColV carriage is likely to be important in this evolutionary trajectory due to the carriage of important virulence genes identified in this analysis. These lineages are unlikely to have extensive chromosomal pathogenicity Islands (otherwise they would NOT be commensal phylogroups). Here, we also sense the misconception from our paper that it was asserted that ‘ColV plasmids are the main driver of virulence’, which we do not believe, as stated above they are clearly important because they carry and transmit these genes in strains that don’t carry them on the chromosome.

We have now modified the text as follow:

“Overall, these data indicate that within the *Escherichia* genus, the sole presence of the ColV plasmids does not explain virulence in our mouse sepsis model, even though aerobactin and *sit* operons can be found rarely as plasmid borne.”

We also discussed the role of ColV plasmids in the light of the epidemiological evidences for their involvement in specific aspects of extraintestinal virulence as well as in the VAG transmission.

Royer et al., provide further critical data that lends weight to the importance of ColV plasmids as vectors of VAGs in the section The HPI needs other VAGs to express full virulence

Line 204 “The avirulent strains despite the presence of the HPI belonged to phylogroups A (n=5/9), B2 203 (n=2/9), C (n=2/9) (Figure S7). Moreover, when we compared strains carrying the HPI depending on the presence of the other operons, the presence of the HPI alone was associated with significantly fewer mice

killed per strain than in the cases of the HPI with *sit*, *sit/iro*, *sit/iuc* or *sit/iuc/iro*. No differences were observed for strains carrying only *iro*, *iuc* or *iro/iuc* probably due to the limited number of strains. Overall, these data indicate that the presence of HPI is necessary but not always sufficient for a strain to be virulent, with other iron uptake systems required to express full virulence. Conversely, these accessory systems alone are not sufficient to acquire full virulence.

Finally, the authors highlight the importance of ColV in ST58 in the following section:

Lines 228-239. “Moreover, these plasmidic VAGs usually occurred in strains which carry *fyuA*, except in the commensalism-associated ST10. More globally, we can see specific patterns. In the case of ST131, the overall pattern is dominated by a very high prevalence and frequent co-occurrence of chromosomal VAGs, with a notable absence of *iroN*. This chromosomal dominant pattern is exacerbated in the ST73 with an almost complete fixation of VAGs on the chromosome. The ST69 and ST95 both present high prevalence and co-occurrence frequencies of chromosomal and plasmidic VAGs, without chromosomal *iroN* or *iucA*, respectively. Of note, this is in agreement with the ColV plasmids frequently observed in the ST95. On the opposite and consistent with its commensal behaviour, the ST10 carries few VAGs, which are usually clustered on plasmids. Finally, for the STc58, in accordance with our previous observation and data from Reid et al., *iroN*, *iucA* and *sitA* co-occurred mainly and frequently on plasmids and are usually associated with the HPI.”

Royer et al., clearly stated that our paper highlighted the co-occurrence of HPI and ColV- which carry *iroN*, *iucA* and *sitA* which is at odds with their concluding statement in the Introduction (‘Using phylogenomic comparative analysis, Reid et al. point to a heightened role of ColV plasmid over the HPI in the evolution of this clone’)

In conclusion, Royer et al., provide evidence that supports a major conclusion from our paper that is encapsulated in the title “A role for ColV plasmids in the evolution of pathogenic *Escherichia coli* ST58” given the lack of chromosomal virulence factors in commensal *E. coli* phylogroup lineages that have emerged to cause serious disease in humans.

Once again, we apologize for the first version of our manuscript that was too centered on a discussion of the Reid et al. paper. We are presenting a more balanced manuscript now.

For clarity Royer et al., may consider highlighting the disparities in the selection of genomes used for their GWAS analysis. It is quite difficult to keep track of which results sections/data were generated on which collection – there are four under analysis:

1. CC87
2. 370 phylogenetically diverse strains
3. 2200 genomes from refseq
4. A collection (n=?) of ST131, ST73, ST69, ST10, ST95 and STc58

The collection referred to in point 4 corresponds to a subset of the Refseq genomes. The number of genomes in each ST is available in Figure 5. We have now also added the number of genomes in each ST in the main text.

“First, we focused on the five main ST responsible for bacteremia in France [ST131 (n=123), ST73 (n=41), ST69 (n=33), ST95 (n=46), ST10 (n=48)] as well as the STc58 (n=37) to quantify co-occurrence frequencies and prevalence of VAG pairs.”

It should be noted that the collection of 370 genomes is 30% B2, 19% A, 11% B1, 10% C and 5% D etc. The collection of ~2200 is 33% A, 21% B1, 15% B2, 13% E and 9% D etc. ColV is likely to play a key role in the

emergence of pathogenic commensal phylogroups that typically lack chromosomal virulence islands but that have acquired HPI so the percentage of B2 in any analysis will cloud interpretation of the importance of plasmidic VAGs.

The point of all this is that the authors might consider revising and nuancing some of their comments about ColV, as their data and ours show that they carry important VAGs that play a role in supplementing the role of HPI as a key virulence island.

Dear Reviewer, we agree with your review, we present now a more balanced manuscript thank to yours and other reviews. Our first version probably reflected a general frustration at not seeing functional tests in many comparative genomics studies.

Reviewer #3 (Remarks to the Author):

Royer and colleagues present a genomic analysis of the evolution of virulence associated genes in *E. coli*, with a special focus on iron acquisition systems. Key to many of the findings was a previously generated compendium of mouse experiments evaluating the outcome of bloodstream infections with ~300 diverse *E. coli* strains. The authors begin by focusing on the emergent ST58 B1 lineage and perform a genome-wide association study to identify genes/variants associated with death of mice in the aforementioned experiments. The only hit identified was the previously described high pathogenicity island (HPI), harboring the yersiniabactin iron scavenging system. The authors made especial note that they did not identify the colV plasmid to be associated with virulence, contrasting/conflicting with a recent report in *Nature Communications* (Reid et al., 2022). The authors then go on to examine the role of virulence associated genes, and col plasmids, more broadly in *E. coli* virulence by performing GWAS in the diverse *E. coli* lineages from the compendium. These GWAS analyses showed a strong signal for the presence of HPI, as well as other iron scavenging systems (*iuc* and *sit*), in a manner that appears dependent on the presence of HPI. The authors further show that these GWAS hits are independent of whether the associated genes are encoded on the chromosome or plasmids. Lastly, the authors examine the co-association of different virulence genes, and show that they were all acquired multiple times independently and display different patterns of co-association in different *E. coli* lineages. The authors use this observation to infer that there is a strong selective pressure to acquire these genes, and that there are likely genetic interactions among them that influence their patterns of association. Overall, this is an interesting analysis, marrying comparative genomics and GWAS to test specific hypotheses regarding the relative roles of different operons in a virulence proxy. However, it is important to note that several of the key observations are more confirmatory of previous publications; association of iron scavenging loci with death in this mouse compendia were previously reported by the authors (Galardini et al., *PLoS Genetics*, 2020) and the observation of HPI being a lineage defining trait for virulent ST58 sub-lineage was reported by Reid et al. in 2022. The two key novel findings are potential interactions among the iron scavenging loci in this mouse model of bloodstream infection, and the rebuttal of the role of colV plasmids in the emergence of pathogenic ST58. The first observation is potentially interesting (although limited to virulence in the context of bloodstream infections) and is a clever use of the existing data set. With regards to the second observation, while I agree that the Reid et al. paper is limited by having no experiments (i.e. just enrichment of genetic loci among exPEc isolates), the current study has its own significant limitations in only focusing on the fitness in the bloodstream (i.e. disregarding the potential role of colV in earlier stages of infection).

Major comments

1. There are a couple of instances where the authors make statements based on Figures, but I did not see statistical support for the assertions. Three in particular were: i) the survival curves in figure 2, ii) difference in

virulence for strains having just HPI versus HPI + another virulence genes in Figure 4 and ii) *iro/iuc/aiu* appearing in the presence of HPI, but not vice versa, in Figure 5. For the first and third point please provide statistical support, and for the second please quantify.

Regarding the survival curves in Figure 2B, we have now added the values of the log rank tests computed to test for differences of survival between the different subgroups and controls.

In the case of Figure 4, to quantify differences we have now computed odds ratios for the status “killer” (i.e. at least 9/10 mice killed) as a function of the presence of the HPI for each VAGs combination. These data are now available on figure 4 and figure S8. We have updated the methods to include these comparisons.

Regarding the statistical supports for the figure 5 and figure S10, we have computed odds ratios based on 2x2 contingency tables for every gene pair in a given ST/STc or phylogroup. This approach enables us to quantify the strength co-occurrence, its direction and its significance. The results of these comparisons are available in Table S4.

2. In the discussion the authors conjecture regarding the implications of gene co-occurrence/exclusion in Figure 6 on potential genetic interactions. Given that the interactions described are primarily among iron scavenging loci, I think the discussion would benefit from expanding on how the varying functions/affinities of these systems may account for their distributions, versus unspecified detrimental interactions that may result in their association/exclusion.

We now discussed the function and specificities of these systems.

3. The visualization of the nucleotide diversity in Figure 6 is quite interesting and indicates that the copies of iron scavenging genes on plasmids trace back to a single mobilization event. It would be interesting to see where the corresponding plasmid associated operons fit onto the chromosomal tree, to gain insight into their origin.

We would like to thank the reviewer for this very interesting suggestion.

As proposed, we performed the phylogenetic analysis on both chromosome and plasmid sequences. Then, we annotated the unrooted trees to highlight the location (chromosome or plasmid), the MLST of the strains and the insertion site in the case of chromosomal VAGs. These trees are now available on figure S12, S13 and S14.

In the case of *iro*, we were able to identify two interesting cases (figure S12). First, we observed a probable mobilization of *iro* sequences from plasmid to the chromosome of ST95 strains. Second, we found a potential mobilization of chromosomal sequences from ST392 strains to plasmids carried by several ST (ST14, ST69, ST657, ST56). Of note and as expected, the chromosomal sequences are more diverse and tend to group by ST whereas for plasmid sequences the branch of the tree are short and mix many different STs.

In the case of *iuc*, we found two main clusters of chromosomal sequences and a third one composed mainly of plasmid sequences (figure S13). Again, we were able to identify potential mobilization of genes from plasmids to chromosome (with many different ST and insertion sites) as well as mobilization from chromosome to plasmid. Of note, chromosomal sequences tend to group by insertion site in agreement with the hypothesis of multiple acquisition events along the evolutionary history of the population.

Finally, for *sit* operon, we also observed a clustering that follow both location, ST and insertion sites (figure S14). We found a probable mobilization event from chromosome to plasmid carried by diverse ST (ST117, ST457 and ST131). However, it is difficult to clearly define the MLST from which the event took place. On the contrary, we observed two cases where plasmid sequences could have been mobilized on the chromosome, each time only once (into an ST131 strain and an ST410 strain).

From these data we can only speculate on how mobilization occurred (chromosome to plasmid or plasmid to chromosome) considering the most parsimonious scenario. Nevertheless, it clearly highlights the exchanges that can occur between plasmids and chromosomes and adds evidence on the selection process acting on these operons. We have added these data to the results and discussion sections.

Reviewer #1 (Remarks to the Author):

This revised manuscript is excellent. The additional analyses add a great deal of value in clarifying the relationship between the iron acquisition systems and virulence. I would point out a few minor points for clarity.

Line 325: HPI gene inactivation is associated (with) the presence of chromosomal *iro* and *sit* genes

Lines 456-459: "Such reductive evolution is the hallmark of specific niche-adapted organisms hardly compatible with the *E. coli* lifestyle alternating between its primary (gut of vertebrates) and secondary (water and sediments) habitat with occasional opportunistic infections^{6,26}."

If this statement is meant to imply that some ExPEC strains/lineages are undergoing niche adaptation, trading overall versatility for increased virulence in specific contexts, then I would heartily agree, and their data seems to support such. Reference 26 cited by the authors has been cited to support the opposite idea, that loss of antivirulence genes is important in evolution of *E. coli* virulence. For instance, Suresh et al. (*mBio* 2021 Jan-Feb; 12(1): e03634-20) who suggest an incompatibility between the *pks* island and some antimicrobial resistance genes in ST95.

Line 463-464: siderophores may be maladaptive in environments in which they are functionally redundant, (and therefore are prone to inactivation)

Line 466: is not involved in our mouse sepsis model

David Erickson

Reviewer #3 (Remarks to the Author):

I thank the authors for their thoughtful response to comments. All of my prior concerns/comments have been addressed.

Reviewer comments and author answers (in red)

Reviewer #1 (Remarks to the Author):

This revised manuscript is excellent. The additional analyses add a great deal of value in clarifying the relationship between the iron acquisition systems and virulence. I would point out a few minor points for clarity.

Thank you

Line 325: HPI gene inactivation is associated (with) the presence of chromosomal iro and sit genes

Done

Lines 456-459: “Such reductive evolution is the hallmark of specific niche-adapted organisms hardly compatible with the *E. coli* lifestyle alternating between its primary (gut of vertebrates) and secondary (water and sediments) habitat with occasional opportunistic infections^{6,26}.”

If this statement is meant to imply that some ExPEC strains/lineages are undergoing niche adaptation, trading overall versatility for increased virulence in specific contexts, then I would heartily agree, and their data seems to support such. Reference 26 cited by the authors has been cited to support the opposite idea, that loss of antivirulence genes is important in evolution of *E. coli* virulence. For instance, Suresh et al. (mBio 2021 Jan-Feb; 12(1): e03634-20) who suggest an incompatibility between the *pks* island and some antimicrobial resistance genes in ST95.

We agree. The paragraph is now as follow:

“The role of the strain genetic background in the acquisition/maintenance of acquired genes has been recently documented for the CRISPR-Cas systems acquired by horizontal gene transfer⁶⁵ by showing inhibition of non-homologous end-joining repair system by Csn2 Cas protein from which encoding genes never co-occurred⁶⁶. It has been also suggested an incompatibility between the *pks* island producing the genotoxin colibactin and some antimicrobial resistance genes in the ST95 (PMID: 33653937). In the same line, antivirulence genes, originally described in *Shigella*, have now been reported in other species as *Salmonella* and *Y. pestis*. These antivirulence genes, located on the recipient genome, must be inactivated for expression of full virulence of horizontally acquired elements⁶⁷. Their functions and the molecular mechanisms at play are very diverse and none involve iron metabolism. Such reductive evolution is the hallmark of highly specific niche-adapted organisms, as the intracellular pathogen and human-restricted *Shigella* spp which belong to the *E. coli* species (PMID: 26923111). However, our data seems to support that some ExPEC strains/lineages are undergoing niche adaptation, trading overall versatility^{6,26} for increased virulence in specific contexts. “

Line 463-464: siderophores may be maladaptive in environments in which they are functionally redundant, (and therefore are prone to inactivation)

Done

Line 466: is not involved in our mouse sepsis model

Done

Reviewer #3 (Remarks to the Author):

I thank the authors for their thoughtful response to comments. All of my prior concerns/comments have been addressed.

Thank you

Erick Denamur